# Evaluating Cross-Modal Reasoning Ability and Problem Characteristics with Multimodal Item Response Theory

**Shunki Uebayashi**[1], **Kento Masui**[2], **Kyohei Atarashi**[1] , **Han Bao**[3,4], **Hisashi Kashima**[1], **Naoto Inoue**[2], **Mayu Otani**[2], **Koh Takeuchi**[1]
[1]Kyoto University [2]CyberAgent. [3]The Institute of Statistical Mathematics [4]Tohoku University

## Abstract

Multimodal Large Language Models (MLLMs) have recently emerged as general architectures capable of reasoning over diverse modalities. Benchmarks for MLLMs should measure their ability for cross-modal integration. However, current benchmarks are filled with shortcut questions, which can be solved using only single modality, and thereby yielding unreliable rankings. For example, in vision-language cases, we can find the correct answer without either the image or the text. These low-quality questions unnecessarily increase the size and computational requirements of benchmarks. We introduce a multi-modal and multidimensional item response theory framework (M3IRT) that extends classical IRT by decomposing both model ability and item difficulty into image-only, text-only, and cross-modal components. M3IRT estimates cross-modal ability of MLLMs and each question's cross-modal difficulty, enabling compact, high-quality subsets that better reflect multimodal reasoning. Across 24 VLMs on three benchmarks, M3IRT prioritizes genuinely cross-modal questions over shortcuts and preserves ranking fidelity even when 50% of items are artificially generated low-quality questions, thereby reducing evaluation cost while improving reliability. M3IRT thus offers a practical tool for assessing cross-modal reasoning and refining multimodal benchmarks.

## 1 Introduction

Multimodal Large Language Models (MLLMs) (Yin et al., 2024) have recently emerged as general architectures capable of reasoning over diverse modalities. A prominent subclass, Visual–Language Models (VLMs), jointly process images and text and are expected to support downstream tasks that require cross-modal reasoning (Jiang & Ye, 2023), such as medical image diagnosis and industrial inspection (Zhang et al., 2024). Consequently, rigorous and trustworthy multimodal benchmarks are essential for practitioners to choose appropriate models (Chen et al., 2024; Yue et al., 2025).

Benchmarks for MLLMs should measure their ability for cross-modal integration. However, current benchmarks are often filled with shortcut questions that can be solved using only single modality (e.g., answerable from text alone or image alone). For example, in vision-language cases, we can find the correct answer without either the image or the text. These low-quality questions unnecessarily increase the size and computational requirements of a benchmark and yields unreliable rankings (Yue et al., 2025). As the pool of candidate models grows, evaluating thousands of mixed-quality questions per model becomes increasingly costly, while single-modality shortcuts further obstacle evaluating the cross-modal reasoning ability.

Item Response Theory (IRT) is a principled framework for assessing subject ability and item difficulty (Fan, 1998). Without knowing the questions and answers, IRT estimates the ability and difficulty as parameters to predict the records of success or failure of a subject on an item. These parameters allow us to construct a compact subset of items tailored to each subject using Computerized Adaptive Testing (CAT) (Weiss & Kingsbury, 1984; Han, 2018). Recent work on LLM has leveraged IRT, where they considered LLM as subject and questions as items, to construct compact and essential subsets of text questions from benchmarks (Polo et al., 2024). However, classical IRT is agnostic

to the modality of inputs and thus contains only a single latent ability or difficulty parameter. IRT cannot determine whether success on a multimodal item reflects true cross-modal reasoning or others.

To address the limitations, we introduce MultiModal and Multidimensional Item Response Theory (M3IRT), and its variant called M2IRT. Our proposed methods simply extend classical IRT by decomposing both model ability and item difficulty into three latent components: image-only, text-only, and cross-modal integration. This decomposition allows us to (i) estimate each VLM's cross-modal ability and (ii) quantify each question's cross-modal difficulty. Using these estimates, our proposed methods identifies genuinely cross-modal items and enables compact, high-quality benchmark subsets that better reflect multimodal reasoning while reducing evaluation cost.

We conduct extensive experiments with 24 VLMs across three benchmarks. We construct semi-synthetic benchmarks by generating simple low-quality questions through the swapping of image or text from the original questions to introduce artificial shortcut or unsolvable questions. We obtain the answers of VLMs and make datasets indicating successes and false. We employ M3IRT, M2IRT, and other methods including IRT to refine our semi-synthetic benchmarks. First, we qualitatively observe that M3IRT prioritizes truly cross-modal items over shortcuts and preserves ranking fidelity even when 50% of the items are replaced with artificially generated low-quality questions. Representative highly and lower cross-modal difficulty items identified by M3IRT are shown in Figure 1.

Second, we conducted experiments to extract subsets of questions from the dataset as a high-quality problem-discovery task. We quantitatively evaluate the degree of ranking reconstruction for VLMs obtained from a small number of subsets of varying sizes, as well as the proportion of simple low-quality questions included in these small subsets. The former enables high performance with fewer items. The results show that our proposed framework nearly reconstructs the original ranking using only a 10% subset across all datasets, while also reducing the proportion of low-quality questions to less than half that of existing methods.

Our contributions[1] are threefold:

1. We propose M3IRT, which explicitly models modality-specific (image-only, text-only) and cross-modal components of both item difficulty and model ability for multimodal evaluation.

2. We show that M3IRT yields compact, high-quality subsets that emphasize cross-modal reasoning and maintain reliable model rankings at substantially reduced computational cost.

3. Through experiments with 24 VLMs across three benchmarks, we demonstrate that M3IRT is robust to large fractions of low-quality items (up to 50%) and provides interpretable characterizations of both benchmarks and models.

## 2 RELATED WORK

Recent VLM evaluation has relied on large, static benchmarks such as MMMU (Yue et al., 2024), MathVista (Lu et al., 2024), SEED-Bench (Li et al., 2024a), EMMA (Hao et al., 2025) and CCHall (Zhang et al., 2025). These efforts shift the center of evaluation toward integration itself rather than isolated unimodal skills. Static expansions such as MMBench (Liu et al., 2024) broaden ability coverage but still exposed to low-quality question contamination and leakage. Several dynamic or live evaluation approaches have emerged such as VLB/FLEX (Yang et al., 2025) proposes to automatically generate both image and text. MAC (Jiang et al., 2025b) and LiveXiv (Shabtay et al., 2025) automatically constructs VQA from current news and papers. While valuable, these benchmarks still exposed to the risk of contaminating low-quality questions, such as shortcuts.

Existing methods for single-modal benchmarks can be categorized into Non-IRT-based and IRT-based approaches. First, Non-IRT-based approaches include question clustering that selects representative questions from clustering results, such as active testing with multi-stage sampling (Huang et al., 2024), tailored benchmark creation (Yuan et al., 2025), LLM predictability exploration (Ye et al., 2023), and anchor points (Vivek et al., 2024). Adaptive sampling dynamically selects questions based on current assessments of a model's performance, including SubLIME (Xu et al., 2024), Dele (Saranathan et al., 2024), and methods that model inter-example dependencies (Li et al., 2024b). FlashEval (Zhao et al.,

---

[1]Our code and data are available at `https://github.com/CyberAgentAILab/M3IRT`.

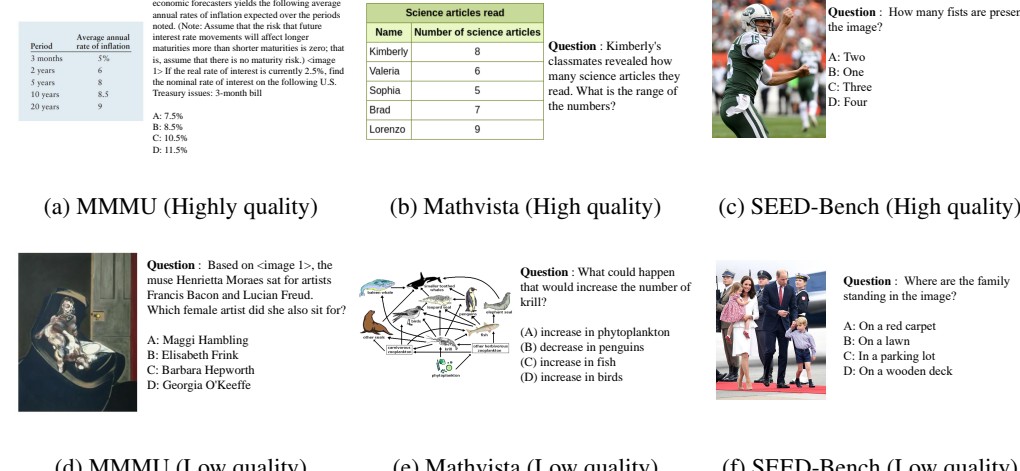

Figure 1: Questions with the highest or lowest cross-modal difficulty $b_j^{\text{cross}}$ detected by M3IRT. Questions with high cross-modal difficulty require both modalities to find the correct answer. However, those with low difficulty allow us to solve using only the image or text.

2024) was recently proposed, offering a novel evolutionary algorithm for text-to-image generation. However, they have not considered whether a question demand the cross-modal integration or not.

Item Response Theory (IRT) (Lord, 1980), originating in psychometrics, provides simultaneous modeling of subject (model) ability and item (question) parameters (e.g., difficulty, discrimination). The application of IRT has expanded to NLP (Lalor et al., 2016), dialogue (Hirai et al., 2023), and recommendation systems (Liu et al., 2023). In the LLM domain, IRT has been leveraged to reduce benchmark volumes; i. e. , MetaBench (Kipnis et al., 2025) distills a sparse benchmark from several benchmarks, and TinyBenchmarks (Polo et al., 2024) provides an efficient cluster-based sampling method. IRT has also been employed for adaptive sampling/testing of LLMs; for example, dynamic test adjustment based on model performance (Zhuang et al., 2023b), CAT-based cognitive ability measurement (Zhuang et al., 2023a), human chatbot evaluation, training of difficulty-calibrated question generators (Jiang et al., 2025a), and automated model evaluation (Guinet et al., 2024).

## 3 BACKGROUND

Consider a collection of MLLMs, treated as subjects and indexed by $M = \{1, \ldots, m\}$, and a multimodal benchmark with questions treated as items and indexed by $N = \{1, \ldots, n\}$. For each subject–question pair $(i, j)$, let $r_{i,j} \in \{0, 1\}$ indicate whether subject $i$ answers question $j$ correctly ($r_{i,j} = 1$) or not ($r_{i,j} = 0$). We denote the resulting response matrix by $R = \{r_{i,j}\}_{(i,j) \in M \times N}$. Our objective is to assess the cross-modal abilities of the MLLMs and the difficulty of the questions, and to identify a compact subset $\hat{N} \subset N$ consisting of items that demand strong cross-modal reasoning.

Item Response Theory (IRT) is a family of latent variable models that jointly infer subject ability and item characteristics from observed response data (Fan, 1998). Given only the pattern of correct or incorrect responses, IRT estimates ability and difficulty parameters and predicts the probability that a subject will answer a given item correctly. We use the two-parameter logistic (2PL) model, which can be viewed as a logistic regression with item-specific slope and threshold:

$$\Pr(r_{i,j} = 1 \mid \theta_i, a_j, b_j) = \sigma\big(a_j(\theta_i - b_j)\big), \tag{1}$$

where $\sigma(x) = 1/(1 + \exp(-x))$ is the sigmoid function. For each subject $i$, we define an ability parameter $\theta_i \in \mathbb{R}$; higher values indicate a greater propensity to answer difficult items correctly. For each item $j$, we define a discrimination parameter $a_j > 0$ and a difficulty parameter $b_j \in \mathbb{R}$. Larger $a_j$ means the probability of a correct response is more sensitive to changes in ability, whereas smaller $a_j$ implies weaker sensitivity. As the difficulty $b_j$ increases, greater ability is required to achieve a

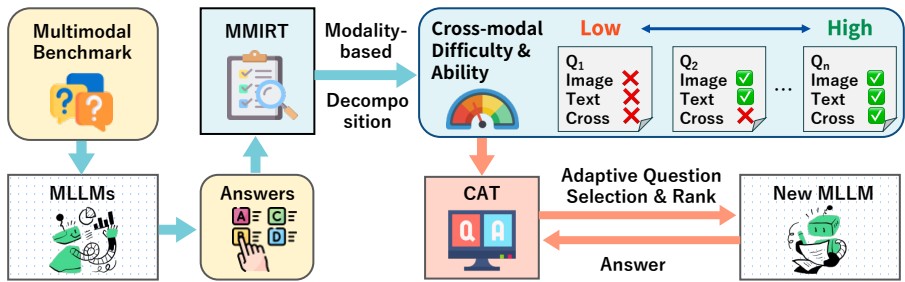

Figure 2: M2IRT investigates the modality-specific and cross-modal difficulties of questions that enables to contract a tailored, compact, and high-quality subset for evaluating a new MLLM.

high probability of a correct response. IRT has been applied to CAT (Weiss & Kingsbury, 1984) to select test questions from an item pool to estimate a subject ability. Namely, we randomly initialize a student ability, select a question with the maximum Fisher information for a current ability, get an answer, and update a subject ability. We repeat this procedure.

Multi-dimensional IRT (MIRT) is a method that extends IRT to consider the relationship between models and questions in a more complex manner (Reckase, 2009). This method supposes a $d$-dimensional latent parameter space. The ability vector for subject $i$ is $\boldsymbol{\theta}_i \in \mathbb{R}^d$, and the difficulty and discriminative vectors for question $j$ as $\boldsymbol{a}_j, \boldsymbol{b}_j \in \mathbb{R}^d$. MIRT parametrizes the probability for providing a correct answer for a pair of $(i, j)$ and finds maximum likelihood estimator:

$$\hat{P}(r_{i,j} = 1) = \sigma\big(\boldsymbol{a}_j^\top \boldsymbol{\theta}_i - \boldsymbol{b}_j\big), \ \hat{P}(r_{i,j} = 0) = 1 - \hat{P}(r_{i,j} = 1). \tag{2}$$

## 4 PROPOSED METHOD

To assess modality-specific and cross-modal properties of MLLMs and multimodal benchmarks, we introduce the decomposition of the standard IRT parameters into latent components. Building on this decomposition, we introduce MultiModal Item Response Theory (M2IRT) and Multidimensional MultiModal Item Response Theory (M3IRT) as extensions of classical IRT and MIRT. We also develop a procedure for selecting a compact subset of benchmark items tailored to these models. Figure 2 illustrates the overall framework. Although the method is applicable to arbitrary modalities (e.g., action, audio), this paper primarily focuses on vision and language.

### 4.1 MODALITY-BASED DECOMPOSITION OF IRT PARAMETERS

We assume that an MLLM has modality-specific abilities as well as an ability to integrate information across modalities. Likewise, each multimodal question exhibits modality-specific and cross-modal characteristics that can determine whether a subject can provide the correct answer.

In the vision–language setting, we define binary indicators $s^{\text{image}}, s^{\text{text}} \in \{0, 1\}$ to represent the modalities present in a question: $s^{\text{image}} = 1$ if an image is provided and $s^{\text{text}} = 1$ if text is provided; otherwise, the indicator is 0. Let $s = (s^{\text{image}}, s^{\text{text}}) \in S = \{(0,0), (0,1), (1,0), (1,1)\}$ denote a format of representing a question. When $(s^{\text{image}}, s^{\text{text}}) = (0, 0)$, the stimulus are withheld and the subject answers using only a guess from introductions or the multiple-choice options.

We assume each subject has a base reasoning ability that, depending on the input format $s$, combines with image-specific, text-specific, and cross-modal integration abilities. For subject $i$, denote the base, image, text, and cross-modal abilities by $\theta_i^{\text{base}}, \theta_i^{\text{image}}, \theta_i^{\text{text}}, \theta_i^{\text{cross}} \in [0, q]$, respectively, where $q \geq 0$ is a shared upper bound that balances their scales. Given a question $j$ and its modality indicator $s$, we define the ability of a subject $j$ as follows:

$$\theta_i(s) = \theta_i^{\text{base}} + s^{\text{image}}\theta_i^{\text{image}} + s^{\text{text}}\theta_i^{\text{text}} + s^{\text{image}}s^{\text{text}}\theta_i^{\text{cross}}. \tag{3}$$

The second and third terms contribute when an image or text is present, respectively; the fourth term contributes only when both are present. This construction naturally extends to additional modalities.

We view answering as exploiting hints provided by the item. For item $j$, let $b_j^{\text{base}}$, $b_j^{\text{image}}$, $b_j^{\text{text}}$, $b_j^{\text{cross}} \in [0, q]$ be the base, image, text, and cross-modal difficulties, respectively, using the same upper bound $q \geq 0$. We define the difficulty $b_j(s)$ of question $j$ given the indicator $s$ as

$$b_j(s) = b_j^{\text{base}} - s^{\text{image}} b_j^{\text{image}} - s^{\text{text}} b_j^{\text{text}} - s^{\text{image}} s^{\text{text}} b_j^{\text{cross}}. \tag{4}$$

Similarly, let $a_j^{\text{base}} \in [0, q]$ be the base discrimination, let $a_j^{\text{image}}, a_j^{\text{text}}, a_j^{\text{cross}} \in [0, q]$ capture the contributions from image, text, and cross-modal integration. The discrimination becomes

$$a_j(s) = a_j^{\text{base}} + s^{\text{image}} a_j^{\text{image}} + s^{\text{text}} a_j^{\text{text}} + s^{\text{image}} s^{\text{text}} a_j^{\text{cross}}. \tag{5}$$

In a general setting, we define indicators to represent the all modalities in a benchmark, and extend parameters applicable to represent combinations of modalities.

## 4.2 MULTIMODAL ITEM RESPONSE THEORY (M2IRT)

To capture cross-modal behavior, we control which modalities are provided, thus each subject answers each item under the four input formats corresponding to all $s \in S$. For each subject–question-format combination $(i, j, s)$, let $r_{i,j,s} \in \{0, 1\}$ indicate whether subject $i$ answers question $j$ given the format indicator $s$ correctly ($r_{i,j,s} = 1$) or not ($r_{i,j,s} = 0$). We denote full response set as the resulting response tensor by $R' = \{r_{i,j,s}\}_{(i,j,j) \in M \times N \times S}$.

M2IRT extends the logistic IRT model in Equation 1. Given discrimination $a_j(s)$, difficulty $b_j(s)$, and ability $\theta_i(s)$, we define $z_{i,j,s} = a_j(s)(\theta_i(s) - b_j(s))$ and introduce M2IRT as follows:

$$\hat{P}(r_{i,j,s} = 1) = \sigma(z_{i,j,s}) \quad \text{and} \quad \hat{P}(r_{i,j,s} = 0) = 1 - \hat{P}(r_{i,j,s} = 1). \tag{6}$$

This parameterization captures the modality-aware behavior of subject $i$ on item $j$.

## 4.3 MULTIMODAL MULTI-DIMENSIONAL ITEM RESPONSE THEORY (M3IRT)

M3IRT extends the logistic MIRT model in Equation 2 with the modality-based decomposition. We modify the decomposed components into vectors. For subject $i$, define the ability vector $\boldsymbol{\theta}_i = [\theta_i^{\text{base}}, \theta_i^{\text{image}}, \theta_i^{\text{text}}, \theta_i^{\text{cross}}]^\top$. For item $j$, define the discrimination and difficulty vectors $\boldsymbol{a}_j = [a_j^{\text{base}}, a_j^{\text{image}}, a_j^{\text{text}}, a_j^{\text{cross}}]^\top$, $\boldsymbol{b}_j = [b_j^{\text{base}}, b_j^{\text{image}}, b_j^{\text{text}}, b_j^{\text{cross}}]^\top$. For convenience, we introduce a format indicator vector $\boldsymbol{s} = [1, -s^{\text{image}}, -s^{\text{text}}, -s^{\text{image}} s^{\text{text}}]^\top$, where the negative signs align with the subtractive role of the modality terms in Equation 4 and with the decomposition in Equation 5. From these vectors, we define $z'_{i,j,s} = \boldsymbol{a}_j^\top \operatorname{diag}(\boldsymbol{s}) \boldsymbol{\theta}_i - \boldsymbol{s}^\top \boldsymbol{b}_j$. We propose M3IRT as follows:

$$\hat{P}(r_{i,j,s} = 1) = \sigma(z'_{i,j,s}) \quad \text{and} \quad \hat{P}(r_{i,j,s} = 0) = 1 - \hat{P}(r_{i,j,s} = 1). \tag{7}$$

Here, $\operatorname{diag}(\boldsymbol{s})$ is the diagonal matrix whose diagonal elements are $\boldsymbol{s}$. The probabilistic model equation 6 is a variant of multi-dimensional IRT with the parametrization $z_{i,j,s}$. This parametrization takes in the modality-aware nature of subject $i$ when answering multimodal question $j$.

## 4.4 LEARNING M3IRT USING STOCHASTIC GRADIENT DESCENT

Instead of the EM algorithm commonly used in IRT, we estimate M3IRT parameters with stochastic gradient descent (SGD). Let a training dataset as $R'' \subset R'$. Given $R''$ and the Bernoulli model in Equation 6, the negative log-likelihood is the negative log likelihood of is

$$\mathcal{L}(\Theta) = -\sum_{(i,j,s) \in R''} \left( r_{i,j,s} \log \hat{P}(r_{i,j,s} = 1) + (1 - r_{i,j,s}) \log \hat{P}(r_{i,j,s} = 0) \right), \tag{8}$$

where the parameters set is $\Theta = \{\{\boldsymbol{a}_j\}_{j \in N}, \{\boldsymbol{b}_j\}_{j \in N}, \{\boldsymbol{\theta}_i\}_{i \in M}\}$. We minimize $\mathcal{L}(\Theta)$ busing mini-bach SGD, $\hat{\Theta} = \operatorname{argmin}_\Theta \mathcal{L}(\Theta)$. We can estimate M2IRT in a similar manner. Note that our approach does not require a dense response matrix: M2IRT and M3IRT can be learned from partially observed data like a tensor completion, reducing the cost of evaluating MLLMs and benchmarks.

### 4.5 Computer Adaptive Test with M2IRT and M3IRT

We integrate M2IRT and M3IRT with classical Computerized Adaptive Testing (CAT) (Weiss & Kingsbury, 1984) to adaptively select an informative subset of items $\hat{N} \subseteq N$, guided by Fisher information. For M2IRT model, the Fisher information of item $j$ for subject $i$ under format $s$ is

$$I_{i,j} = \hat{P}(r_{i,j,s} = 1)\hat{P}(r_{i,j,s} = 0)(a_j(s))^2, \tag{9}$$

where $\hat{P}(r_{i,j,s} = 1)$ is given by Equation 6. For the multidimensional M3IRT model, the Fisher information matrix for item $j$ at ability $\theta$ is

$$\boldsymbol{I}_{i,j} = \hat{P}(r_{i,j,s} = 1)\hat{P}(r_{i,j,s} = 0)(\mathrm{diag}(\boldsymbol{s})\boldsymbol{a}_j)(\mathrm{diag}(\boldsymbol{s})\boldsymbol{a}_j)^\top. \tag{10}$$

We adopt the D-optimality criterion (Mulder & Linden, 2009) to minimize estimation uncertainty by maximizing the determinant of the cumulative information. Let $U_i \subseteq N$ be the set of items not yet answered by subject $i$. At stage $t$, given the cumulative information matrix $\boldsymbol{I}_i^{(t-1)}$, we select the next item and update:

$$j^* = \underset{j \in U_i}{\mathrm{argmax}} \det\left(\boldsymbol{I}_i^{(t-1)} + \boldsymbol{I}_{ij}\right), \quad \boldsymbol{I}_i^{(t)} = \boldsymbol{I}_i^{(t-1)} + \boldsymbol{I}_{ij^*}. \tag{11}$$

Iterating this rule yields a subset that is maximally informative for estimating the subject's ability.

## 5 Experiment

### 5.1 Datasets and Baselines

We employed three benchmarks for VLMs in this experiment. **MMMU** (Yue et al., 2024) is designed to evaluate the reasoning capabilities of VLM through undergraduate-level questions in diverse disciplines such as art and design, business, and science. We used 900 questions in the validation set. **MathVista** (Lu et al., 2024) evaluates mathematical reasoning capabilities through questions involving visual context including puzzle figures and graphs. We used 1000 questions of the test-min set. **SEED-Bench** (Li et al., 2024a) is a large-scale benchmark designed to comprehensively evaluate the multimodal abilities. We used 1000 questions from **L1** and **L2** sets.

To simulate the presence of questionable samples in real-world datasets, we constructed a synthetically contaminated benchmark. We made semi-synthetic benchmarks by generating simple low-quality questions through the swapping of image or text from the original questions. This process introduces artificial shortcut or unsolvable questions. We compile a benchmark contaminated with 50% low-quality questions. We provide a detailed description of our data generation process in Appendix A. To create more realistic low-quality questions, methods such as modifying text and options using LLM or adding noise to images could be considered. Since such methods make the experiment overly complex, we excluded them. Note that our method learns ability and problem characteristics from whether VLMs answer questions correctly, even if there are different types of low-quality questions, the estimation results are unlikely to change.

We collected responses from 24 VLMs, including the GPT-4.1 series, Gemini-2.0 series, and Claude-3.7 series, as well as open-source models such as Qwen-2.5-vl (Bai et al., 2025), Llama-3.2 (Meta, 2024), and Pixtral (Agrawal et al., 2024). On SEED-Bench, since Claude-sonnet-3 became unavailable at the start of the experiments on SEED-Bench, the experiments on SEED-Bench were conducted with 23 models other than Claude-sonnet-3.

We use four baseline methods in our experiments. **Random** selects subset questions at random. **IRT** uses a Fisher information-based subset selection estimated by IRT (Reckase, 2009). **MIRT** uses a Fisher information-matrix-based subset selection estimated by MIRT (Reckase, 2009). **TinyBenchmarks** (Polo et al., 2024) is an IRT-based problem selection method for benchmark refinement in LLM. **FlashEval** (Zhao et al., 2024) is a SOTA to select prompts for image generation. We extended FlashEval to deal with VLM benchmarks by regarding questions as prompts.

We implemented our proposed method with PyTorch (Paszke, 2019), and used Adam optimizer (Kingma & Ba, 2014) whose learning rate was 0.01. We used a grid search to select hyperparameter $q$ from $2, 4, 8, 16$. We selected the optimal hyperparamters based on the highest AUC in predicting the correctness of the VLMs' responses on the validation dataset. We provide the detailed explanation of the experimental setting in Appendix E.

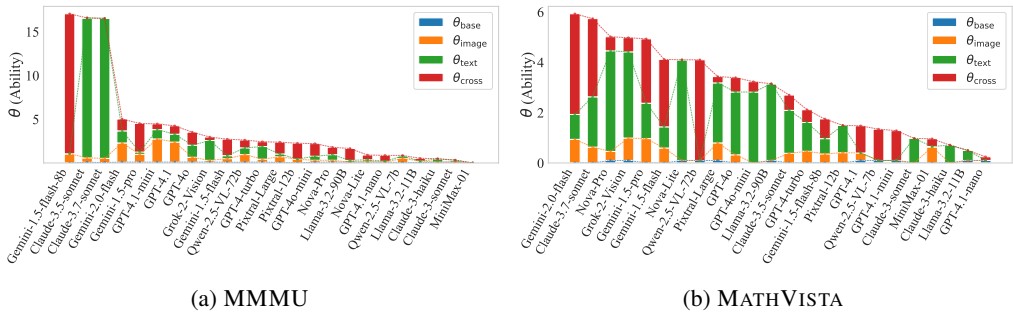

(a) MMMU

(b) MATHVISTA

Figure 3: Distributions of $\theta$ estimated by M3IRT sorted in descending order.

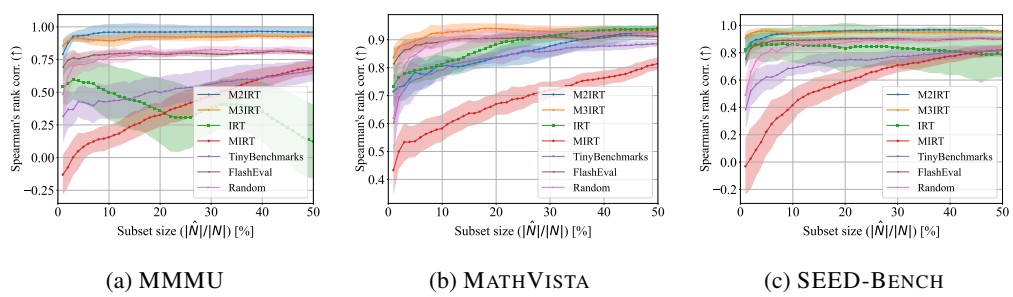

(a) MMMU

(b) MATHVISTA

(c) SEED-BENCH

Figure 4: The average and standard deviation of Spearman's rank correlations between model rankings on the original benchmark and those estimated on extracted question subsets with different sizes.

## 5.2 MULTIMODAL DIFFICULTY AND ABILITY DECOMPOSITION

M3IRT estimates the extent to which a question requires cross-modal reasoning, represented by difficulty $b_j^{\text{cross}}$. This facilitates the identification of questions that truly benefit model's cross-modal capability assessment. Figure 1 shows examples of questions with high and low $b_j^{\text{cross}}$. The questions with low $b_j^{\text{cross}}$ are judged that they can be solved only with images or text. For example, the bottom one in MMMU can be answered based on knowledge of artists without looking into the image. On the other hand, the questions with high $b_j^{\text{cross}}$ cannot be solved if either the image or the text is missing. For example, the one in MATHVISTA requires reading the numerical values in the table that cannot be confirmed only by the question text. Similarly, if only images are provided, it is not clear what is being asked about in the table. We provide more examples in Appendix C.

M3IRT also estimates the extent to which the reasoning ability for each modality contributes to the VLM performance. Figure 3 shows the decomposed reasoning abilities of VLMs. The top-performing model on MMMU exhibits high ($\theta_i^{\text{cross}}$), suggesting strong cross-modal reasoning capabilities. On the other hand, the second and third best-performing models demonstrate high textual reasoning ability ($\theta_i^{\text{text}}$) but limited cross-modal reasoning capability. This analysis suggests that these latter VLMs rely heavily on text understanding when solving the MMMU benchmark, rather than effectively integrating visual information. In MATHVISTA, most VLMs have high $\theta_i^{\text{text}}$. This may reflect MATHVISTA's emphasis on text understanding. Most VLMs also exhibit moderate $\theta_i^{\text{image}}$, suggesting that they also leverage the visual ability to process diagrams and graphs. The result for SEED-BENCH is shown Appendix C.

## 5.3 BENCHMARK REFINEMENT

We investigate whether a method can extract a compact subset of questions that enables us to evaluate the performance of unseen VLMs. We randomly select a VLM from a collection of VLMs and construct a subset of the responses of remaining VLMs. For a method, we select a subset of questions, estimate the performance of the VLM from its responses to the subset, and obtain an estimated ranking of VLMs. We compare the difference between rankings on the original benchmark for all models. We also investigate how much the artificial low-quality questions are included in the subset.

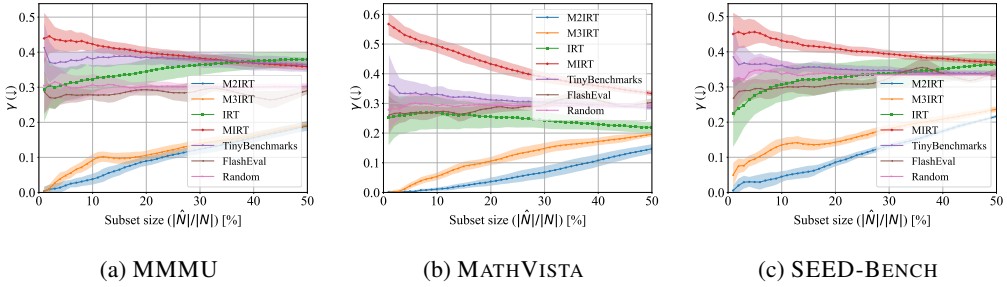

|   |   |   |
|---|---|---|
| (a) MMMU | (b) MATHVISTA | (c) SEED-BENCH |

Figure 5: The average and standard deviation of the proportions of the low-quality questions in extracted question subsets $\gamma$ with different sizes.

We use two measures to assess the quality of a subset $\hat{N} \subseteq N$ selected by a method. First, we assess how much a method avoid the low-quality questions in the estimation of model rankings. We compute the Spearman's rank correlation between model rankings on the original benchmark and the extracted subset. Second, we evaluate how a method can distinguish between the original and low-quality questions. We measure the proportion of low-quality questions in the extracted subset as

$$\gamma = \frac{|\{q \in \hat{N} \mid q \text{ is a low-quality question}\}|}{|\hat{N}|}.$$

We varied the subset size from 1% to 50% of the whole benchmark in 1% increments. We employed CAT with M2IRT using the maximum Fisher information in Sec. 4.5 and M3IRT using D-Optimality. We obtained the average and standard deviation from twenty four independent experiments.

Figure 4 shows the Spearman's rank correlations between the model rankings on the original benchmark and on different sizes of subsets. Figure 5 shows the proportion $\gamma$ with varying size of subsets.

As shown in Figure 4, our methods accurately estimate model rankings from contaminated benchmarks, even with small subsets. In MMMU, M2IRT achieves a rank correlation of 0.9 using only 3% of the benchmark subset, and M3IRT suprizingly achieves a rank correlation of 0.8 using the only 1% subset. FlashEval, which is SOTA but does not account for the presence of low-quality questions, performs similarly to Random. In MATHVISTA, M3IRT achieves a rank correlation of 0.84 with a subset fraction of only 2%, requiring 30% to achieve a rank correlation of 0.9. In SEED-BENCH, M2IRT achieves a rank correlation of 0.9 using only 3% of the benchmark subset, while M3IRT achieves the same rank correlation using only 1% of the benchmark subset.

From Figure 5, we confirmed that the proportion of artificial low-quality questions included in the subset selected by the proposed method is significantly smaller compared to existing methods. In MMMU, even with an extraction subset size of 50%, all proposed methods keep the proportion of low-quality questions notably low at 24%. In contrast, the baseline methods choose substantially more low-quality questions than ours, which skew the estimated model rankings. When extracting 30% of MATHVISTA, the rank correlation between M3IRT and Random is about the same, but $\gamma$ is smaller for M3IRT. We observed similar trends in the results of SEED-BENCH.

## 5.4 ROBUSTNESS AGAINST LOW-QUALITY QUESTIONS

We have evaluated the performance of the proposed method using a subset of questions. Here, we assess its performance as a latent variable method for predicting missing responses from observed ones. First, from the set of questions $N$, we randomly select 100 or 10% questions each for validation and testing, using the remainder as training data. Next, we perform parameter estimation using the training data for both the proposed method and IRT. Finally, we evaluate the prediction performance on the test data using the estimated parameters with ROC-AUC. We measured ROC-AUC by varying the proportion of low-quality problems introduced in Sec. 5.1. We used IRT as a baseline in this experiment. We obtained the average and standard deviation from ten independent experiments.

We show the results in Figure 6. Our proposed methods achieved performance comparable to the standard IRT on ROC-AUC. M2IRT was slightly better than IRT on MMMU, and comparable to IRT

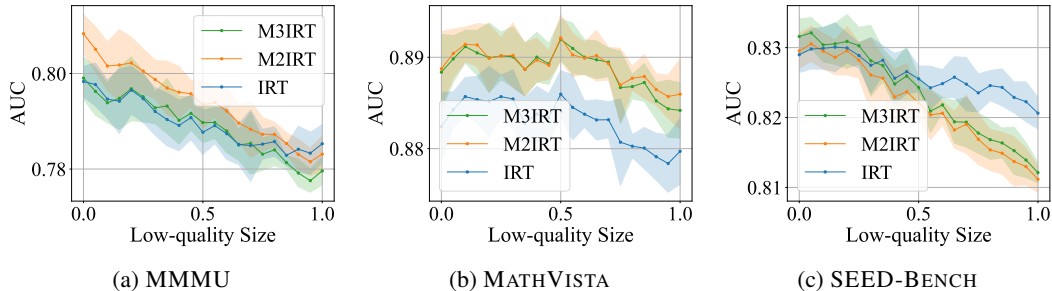

Figure 6: ROC-AUC on predicting answers of the noisy benchmarks containing the different size of the low-quality questions inserted into the original benchmark.

on MATHVISTA and SEED-BENCH. M3IRT was slightly lower than IRT on MATHVISTA but the difference is small. Even when low-quality questions are mixed in, the proposed method and IRT achieve ROC-AUC values around 0.8, suggesting that they effectively capture both the abilites of VLMs and the characteristics of the questions.

## 6 CONCLUSION

We addressed the challenge of assessing cross-modal reasoning characteristics in MLLMs and multimodal benchmarks while reducing evaluation cost. We introduced M3IRT and its variant M2IRT, which decompose both model ability and item difficulty of IRT into image-only, text-only, and cross-modal components. This decomposition enables the identification of highly cross-modal items that require cross-modal reasoning and supports lightweight assessment with far fewer items.

Across three benchmarks and 24 VLMs, we qualitatively evaluated that M3IRT can estimate the degree to which an item requires cross-modal reasoning, and assigns higher cross-modal difficulty to genuinely cross-modal items than to single-modality shortcut. Moreover, analyses with synthetically contaminated benchmarks confirmed that M3IRT and M2IRT yields evaluations aligned with the original benchmarks, demonstrating robustness to low-quality contamination.

**Limitations and future work.** Our study focuses on multiple-choice, which is a typical form of closed-ended questions. Extending the framework to open-ended settings with open-ended questions is a natural next step, enabling the discovery of items that demand stronger cross-modal reasoning and the evaluation of MLLMs under generative outputs. Beyond vision–language, applying the approach to additional modalities (e.g., audio, actions) and developing question-generation methods that control cross-modal difficulty are promising directions.

### ETHICS STATEMENT

This work adheres to the ICLR Code of Ethics. All datasets used in this paper (MMMU, MathVista, SEED-Bench) are publicly released benchmarks, and we strictly followed their respective licenses.

### ACKNOWLEDGMENTS

This work was supported by JST FOREST Program Grant Number JPMJFR232S, JST CREST Grant Number JPMJCR21D1, JST BOOST Grant Number JPMJBY24E8.

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

## A  DETAILS OF LOW-QUALITY QUESTION GENERATION

For MMMU, we made three types of low-quality questions: (A) 300 questions consisting of the image, text, and multiple-choice selected from all different questions, (B) 300 questions where the image was replaced with that from different questions; and (C) 300 questions where the text was replaced with that from different questions. For MATHVISTA, we made 333 questions each for (A), (B), and (C). For SEED-BENCH, we made 333 questions each for (A), (B), and (C).

## B  ALIGNMENT WITH HUMAN ANSWERING PATTERNS

To investigate whether the cross-modal difficulty $b_j^{\text{cross}}$ estimated by M3IRT corresponds to human answering patterns, we conducted a crowdsourcing experiment. We randomly sampled 200 questions per benchmark. Nine crowd workers answered each question under three input formats: Image+Text, Image-only, and Text-only. For each question $j$, we computed the accuracy under each format and defined two measures of cross-modal synergy:

$$\text{Decline}_{\text{mean},j} = \text{Acc}_{\text{image+text},j} - \frac{\text{Acc}_{\text{image},j} + \text{Acc}_{\text{text},j}}{2}, \tag{12}$$

$$\text{Decline}_{\text{max},j} = \text{Acc}_{\text{image+text},j} - \max(\text{Acc}_{\text{image},j}, \text{Acc}_{\text{text},j}). \tag{13}$$

$\text{Decline}_{\text{mean}}$ measures how much the joint presentation improves over the average single-modality performance, while $\text{Decline}_{\text{max}}$ measures the improvement over the best single modality. We report Spearman's rank correlation between $b_j^{\text{cross}}$ and each measure. Inter-annotator agreement is assessed by Fleiss' $\kappa$, where values above 0.6 are generally considered acceptable.

Table 1 summarizes the results. On SEED-BENCH, we observed weak positive correlations between $b_j^{\text{cross}}$ and both $\text{Decline}_{\text{mean}}$ (0.35) and $\text{Decline}_{\text{max}}$ (0.35), with moderate inter-annotator agreement ($\kappa = 0.44$) and crowd worker accuracy of 0.62 (compared to 0.57 for VLMs). On MMMU and MATHVISTA, the correlations were weak or negative ($\text{Decline}_{\text{mean}}$: 0.077 and $-0.16$; $\text{Decline}_{\text{max}}$: 0.090 and $-0.15$, respectively), with low inter-annotator agreement ($\kappa = 0.25$ and 0.28). Crowd worker accuracy on these benchmarks was notably low ($\text{Acc}_{\text{image+text}} = 0.41$ and 0.23), whereas VLMs achieved 0.52 on both. These results indicate that VLMs and humans exhibit different cross-modal answering patterns: the questions that M3IRT identifies as requiring strong cross-modal reasoning for VLMs do not necessarily pose the same cross-modal demand for human solvers. This difference was observed across all three benchmarks, though to varying degrees, and was larger on MMMU and MATHVISTA, where crowd worker accuracy was notably low.

For the human evaluation, we collected responses from nine crowd workers via Amazon Mechanical Turk (MTurk). The tasks consisted solely of answering multiple-choice questions drawn from the aforementioned public benchmarks. No personally identifiable or sensitive information was collected.

Table 1: Relationships between VLMs and human answer patterns

|  | MMMU | MathVista | SEED-Bench |
| --- | --- | --- | --- |
| $\text{Decline}_{max}$ | 0.032 | -0.23 | 0.18 |
| Corr. between $\text{Decline}_{max}$ and $b_j^{\text{cross}}$ | 0.090 | -0.15 | 0.35 |
| $\text{Decline}_{mean}$ | 0.14 | -0.11 | 0.31 |
| Corr. between $\text{Decline}_{mean}$ and $b_j^{\text{cross}}$ | 0.077 | -0.16 | 0.35 |
| Fleiss' $\kappa$ | 0.25 | 0.28 | 0.44 |
| Accuracy of Crowd workers | 0.41 | 0.23 | 0.62 |
| Accuracy of VLMs | 0.52 | 0.52 | 0.57 |

## C  OMITTED RESULTS OF MULTIMODAL DIFFICULTY AND ABILITY DECOMPOSITION

Figure 7 shows the decomposed reasoning abilities of VLMs estimated by M2IRT. Figure 7a, Fig. 7b, and Fig. 7c show the decomposed reasoning abilities of VLMs for MMMU, MATHVISTA, and

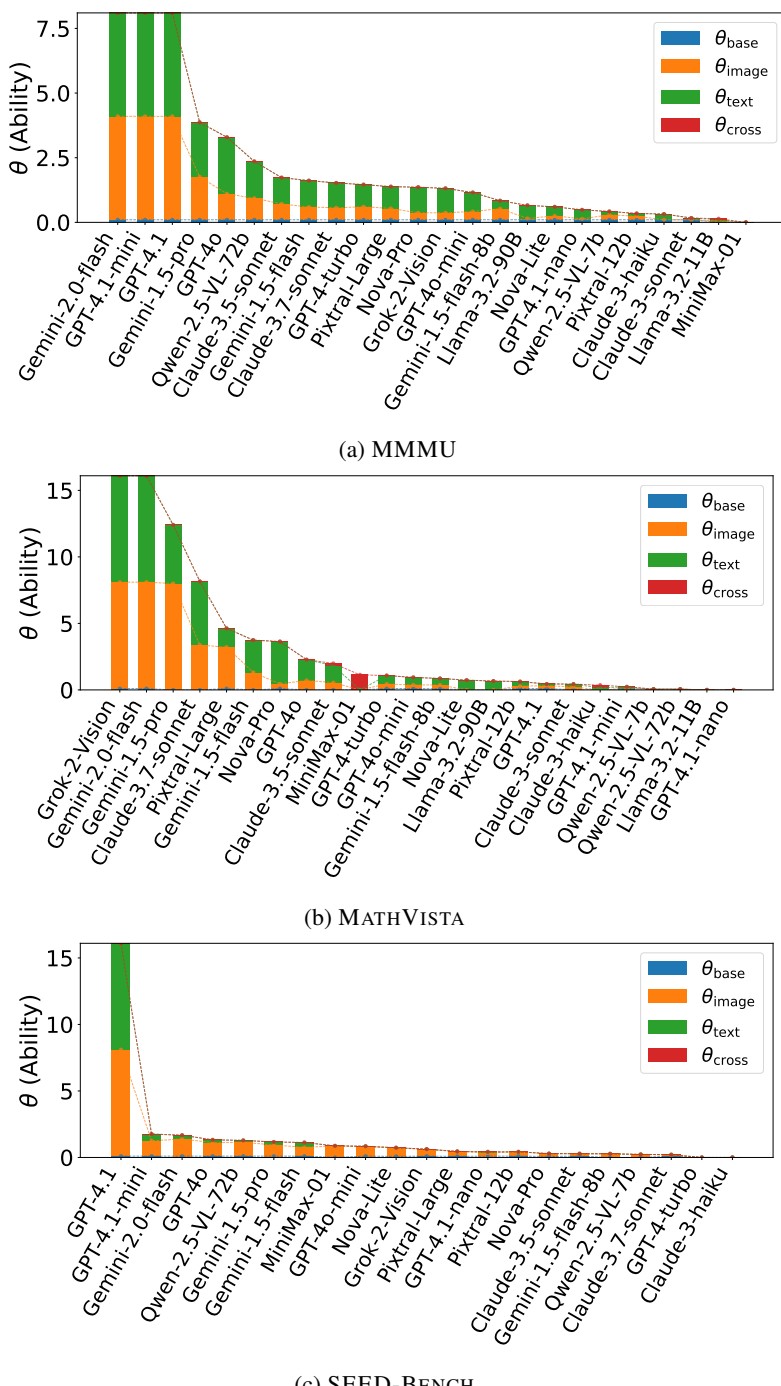

(a) MMMU

(b) MathVista

(c) SEED-Bench

Figure 7: Distributions of $\theta$ estimated by M2IRT sorted in descending order.

SEED-Bench. Figure 8 shows the decomposed reasoning abilities of VLMs for SEED-Bench. In SEED-Bench, most VLMs have high $\theta_i^{\mathrm{image}}$. This result corresponds to the fact that SEED-Bench contains problems which require images strongly to solve.

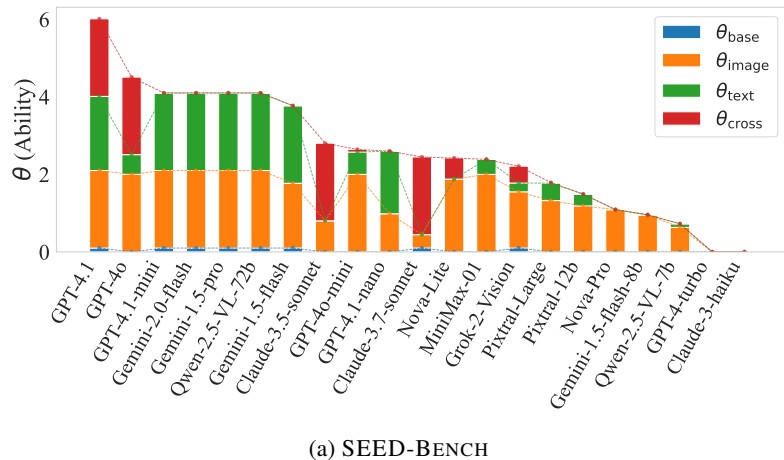

(a) SEED-BENCH

Figure 8: Distributions of $\theta$ estimated by M3IRT sorted in descending order.

## C.1 QUESTIONS WITH HIGH OR LOW CROSS-MODAL DIFFICULTY

We show the questions detected by M2IRT that require the high or low cross-modal reasoning ability from MMMU in Fig. 10 and Fig. 9, from MATHVISTA in Fig. 12 and Fig. 11, and from SEED-BENCH in Fig. 14 and Fig. 13, respectively.

As shown in Fig. 9, Fig. 11, and Fig. 13, questions which require the high cross-modal reasoning ability, whereas questions in Fig. 10, Fig. 12, and Fig. 14 can be solved by using a single-modality only. For example, the question shown in Fig. 10c presents an image of cholera bacteria, where the correct answer (A) can be identified solely from the image and answer choices, even without the text. The question shown in Fig. 12b can be solved correctly without the image if one knows the number of veins for each plant. For the question shown in Fig. 14c , since the question in Fig. 14c, this problem can be solved correctly simply by answering the characters shown in the image. On the other hand, the question shown in Fig. 9b requires both the image, which provides velocity information, and the text, which specifies the particular conditions to identify within the figure. Consequently, the problem cannot be solved correctly if either the image or text component is missing. The question shown in Fig. 11c. The question in Fig. 13c, requires both the image, which provides there is one person who wears black clothes, and the text, which specifies specifying what to count within the figure. Consequently, the problem cannot be solved correctly if either the image or text component is missing. Thus, the $b_j^{\mathrm{cross}}$ successfully distinguishes between questions suitable for evaluating cross-modal ability where essential information is distributed across both image and text, requiring an examination of both to obtain the necessary information and those that do not effectively evaluate cross-modal ability.

| Temperature | 36.8°C (98.2°F) |
| Blood pressure | 140/90 mmHg |
| Heart rate | 105/min |
| Oxygen saturation (at rest) | 92% on room air |
| Oxygen saturation (walking) | 84% on room air |

**Question** : An 82-year-old woman presents to the office with a 1-year history of worsening cough and shortness of breath. She has a 45 pack-year history of cigarette smoking and quit smoking 15 years ago. Vital signs reveal: <image 1>. ECG findings are normal. Her FEV1/FVC ratio is 65% of predicted. The most appropriate inhaled medication for this patient works by blocking which of the following receptors?

A: $\yen$beta $1-adrenergic receptors
B: glucocorticoid receptors
C: histamine H1 receptors
D: leukotriene receptors
E: muscarinic receptors

(a) validation Pharmacy 2

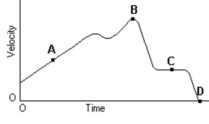

<Image 1>

**Question** : <image 1>Given the graph of the velocity vs. time of a duck flying due south for the winter. At what point did the duck stop its forward motion?

A: A
B: B
C: C
D: D

(b) validation Physics 26

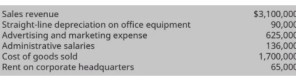

<Image 1>

**Question** : <image 1> Hicks Products produces and sells patio furniture through a national dealership network. They purchase raw materials from a variety of suppliers and all manufacturing, and assembly work is performed at their plant outside of Cleveland, Ohio. They recorded these costs for the year ending December 31, 2017. What is total revenue?

A: $3,100,000
B: $2,616,000
C: $2,474,000
D: $484,000

(c) validation Accounting 12

Figure 9: MMMU: Questions with the high cross-modal difficulties $b_j^{\text{cross}}$

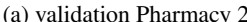

<Image 1>

**Question** : Refer to the figure <image 1>, which term refers to lines that give the impression of calm and tranquility, such as those seen in the ocean and open prairies?

A: Diagonal line
B: Horizontal line
C: Vertical line
D: List spacing

(a) validation Literature 6

<Image 1>

**Question** : Name the written-out ornament, which is marked with bracket. <image 1>

A: acciaccatura
B: appoggiat
C: lower morden
D: upper turns

(b) validation Music 11

<Image 1>

**Question** : The circular rings of muscle that are at the entrance and exit of the stomach are called. Choosing the matching term:<image 1>

A: Cholera
B: Emulsification
C: Anthrax
D: Peristalsis

(c) validation Agriculture 12

Figure 10: MMMU: Questions with the low cross-modal difficulties $b_j^{\text{cross}}$.

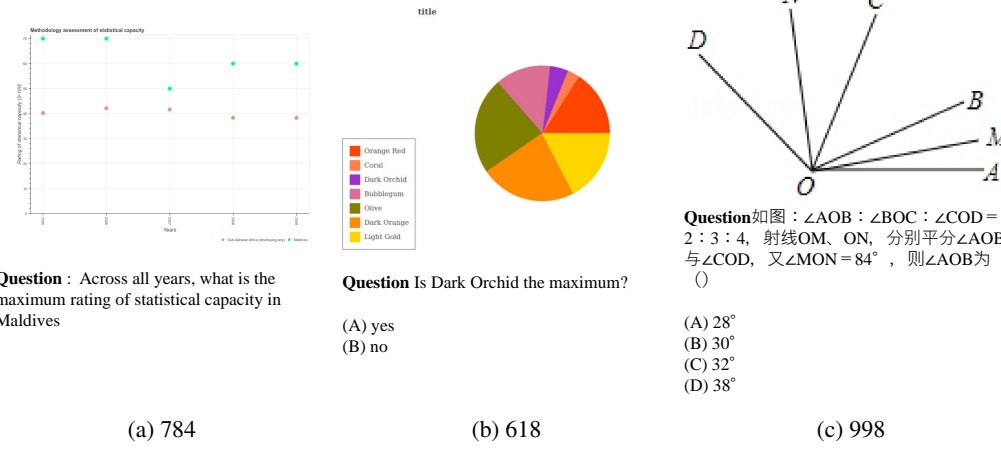

**Question** : Across all years, what is the maximum rating of statistical capacity in Maldives

**Question** Is Dark Orchid the maximum?

(A) yes
(B) no

**Question**如图：∠AOB：∠BOC：∠COD＝2：3：4，射线OM、ON，分别平分∠AOB与∠COD，又∠MON＝84°，则∠AOB为（）

(A) 28°
(B) 30°
(C) 32°
(D) 38°

(a) 784      (b) 618      (c) 998

Figure 11: MATHVISTA: Questions with the high cross-modal difficulties $b_j^{\text{cross}}$. Each caption is the ID of the problem in MATHVISTA

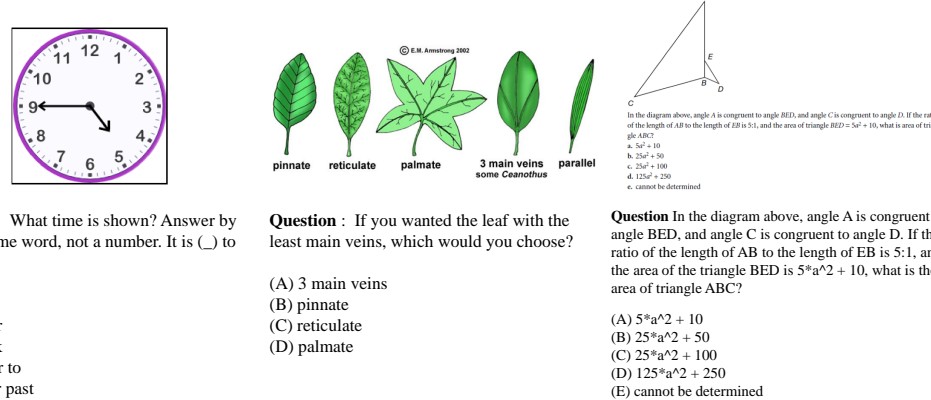

**Question** : What time is shown? Answer by typing a time word, not a number. It is (_) to five.

(A) half
(B) quarter
(C) o'clock
(D) quarter to
(E) quarter past

(a) 531

**Question** : If you wanted the leaf with the least main veins, which would you choose?

(A) 3 main veins
(B) pinnate
(C) reticulate
(D) palmate

(b) 514

**Question** In the diagram above, angle A is congruent to angle BED, and angle C is congruent to angle D. If the ratio of the length of AB to the length of EB is 5:1, and the area of the triangle BED is 5*a^2 + 10, what is the area of triangle ABC?

(A) 5*a^2 + 10
(B) 25*a^2 + 50
(C) 25*a^2 + 100
(D) 125*a^2 + 250
(E) cannot be determined

(c) 315

Figure 12: MATHVISTA: Questions with the low cross-modal difficulties $b_j^{\text{cross}}$. Each caption is the ID of the problem in MATHVISTA

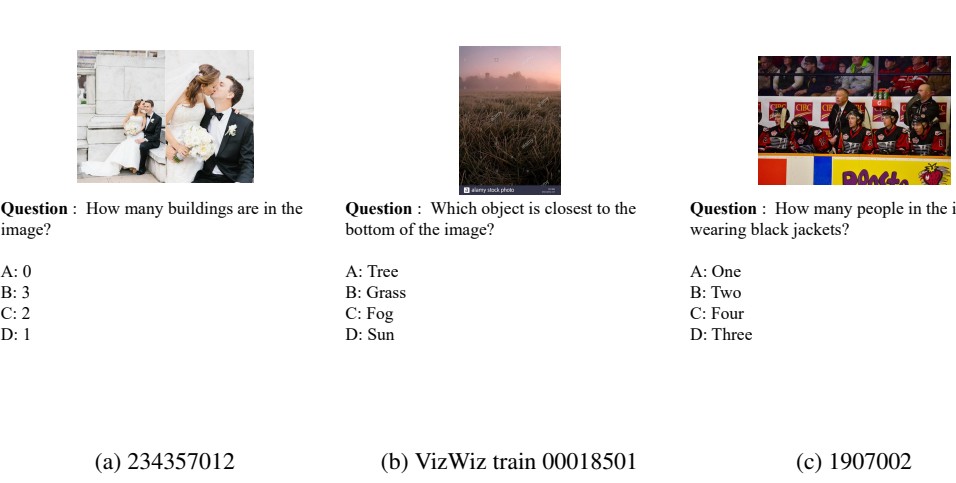

**Question** : How many buildings are in the image?

A: 0
B: 3
C: 2
D: 1

(a) 234357012

**Question** : Which object is closest to the bottom of the image?

A: Tree
B: Grass
C: Fog
D: Sun

(b) VizWiz train 00018501

**Question** : How many people in the image are wearing black jackets?

A: One
B: Two
C: Four
D: Three

(c) 1907002

Figure 13: SEED-BENCH: Questions with the high cross-modal difficulties $b_j^{\text{cross}}$.

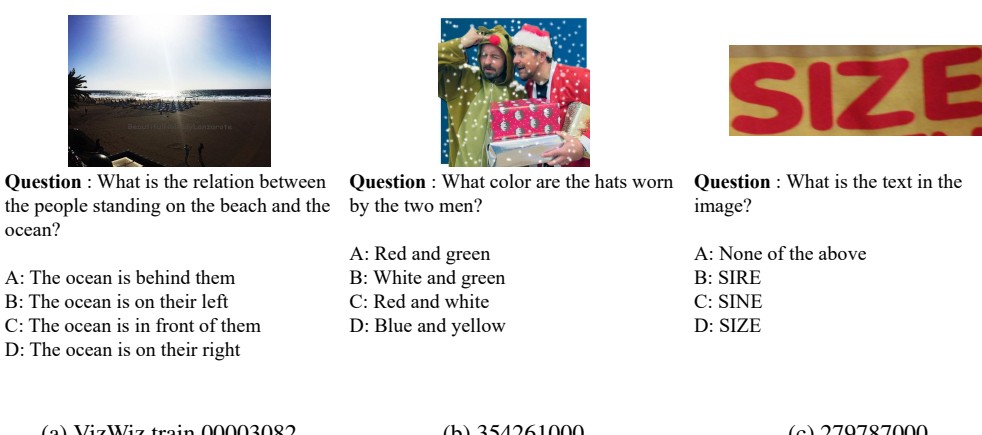

**Question** : What is the relation between the people standing on the beach and the ocean?

A: The ocean is behind them
B: The ocean is on their left
C: The ocean is in front of them
D: The ocean is on their right

(a) VizWiz train 00003082

**Question** : What color are the hats worn by the two men?

A: Red and green
B: White and green
C: Red and white
D: Blue and yellow

(b) 354261000

**Question** : What is the text in the image?

A: None of the above
B: SIRE
C: SINE
D: SIZE

(c) 279787000

Figure 14: VQAAT: Questions with the low cross-modal difficulties $b_j^{\text{cross}}$

Table 3: Estimated $\theta$ on MATHVISTA

| model | $\theta_{\text{base}}$ | $\theta_{\text{image}}$ | $\theta_{\text{text}}$ | $\theta_{\text{cross}}$ | total |
|---|---|---|---|---|---|
| Gemini-2.0-flash | 0.00 | 0.93 | 0.99 | 4.0 | 5.9 |
| Claude-3.7-sonnet | 0.00 | 0.63 | 2.0 | 3.1 | 5.7 |
| Nova-Pro | 0.10 | 0.36 | 4.0 | 0.56 | 5.0 |
| Grok-2-Vision | 0.10 | 0.89 | 3.4 | 0.57 | 5.0 |
| Gemini-1.5-pro | 0.00 | 0.97 | 1.4 | 2.6 | 4.9 |
| Gemini-1.5-flash | 0.00 | 0.60 | 0.84 | 2.7 | 4.1 |
| Nova-Lite | 0.10 | 0.00 | 4.0 | 0.00 | 4.1 |
| Qwen-2.5-VL-72b | 0.10 | 0.00 | 0.00 | 4.0 | 4.1 |
| Pixtral-Large | 0.10 | 0.70 | 2.4 | 0.25 | 3.4 |
| GPT-4o | 0.00 | 0.32 | 2.5 | 0.57 | 3.4 |
| GPT-4o-mini | 0.00 | 0.00 | 2.8 | 0.41 | 3.2 |
| Llama-3.2-90B | 0.10 | 0.00 | 3.0 | 0.00 | 3.1 |
| Claude-3.5-sonnet | 0.00 | 0.38 | 1.7 | 0.61 | 2.7 |
| GPT-4-turbo | 0.00 | 0.46 | 1.1 | 0.51 | 2.1 |
| Gemini-1.5-flash-8b | 0.00 | 0.36 | 0.60 | 0.78 | 1.7 |
| Pixtral-12b | 0.00 | 0.41 | 1.1 | 0.00 | 1.5 |
| GPT-4.1 | 0.10 | 0.30 | 0.00 | 1.1 | 1.5 |
| Qwen-2.5-VL-7b | 0.10 | 0.00 | 0.00 | 1.2 | 1.3 |
| GPT-4.1-mini | 0.10 | 0.00 | 0.00 | 1.2 | 1.3 |
| Claude-3-sonnet | 0.00 | 0.00 | 0.98 | 0.00 | 0.98 |
| MiniMax-01 | 0.00 | 0.64 | 0.02 | 0.31 | 0.96 |
| Claude-3-haiku | 0.00 | 0.00 | 0.71 | 0.00 | 0.71 |
| Llama-3.2-11B | 0.10 | 0.00 | 0.40 | 0.00 | 0.50 |
| GPT-4.1-nano | 0.10 | 0.00 | 0.00 | 0.15 | 0.25 |

## C.2 DETAILED RESULT

Table 2, Table 3, and Table 4 show the values of $\theta$ predicted in Fig. 3 and Fig. 8.

Table 2: Estimated $\theta$ on MMMU

| model | $\theta_{\text{base}}$ | $\theta_{\text{image}}$ | $\theta_{\text{text}}$ | $\theta_{\text{cross}}$ | total |
|---|---|---|---|---|---|
| Gemini-1.5-flash-8b | 0.00 | 1.2 | 0.00 | 4.0 | 5.2 |
| Claude-3.7-sonnet | 0.03 | 0.78 | 4.0 | 0.00 | 4.8 |
| Claude-3.5-sonnet | 0.06 | 0.68 | 4.0 | 0.00 | 4.7 |
| Gemini-2.0-flash | 0.09 | 1.4 | 0.68 | 2.1 | 4.3 |
| Gemini-1.5-pro | 0.10 | 1.0 | 0.29 | 2.8 | 4.2 |
| GPT-4.1-mini | 0.09 | 1.8 | 0.70 | 1.4 | 3.9 |
| Pixtrl-large | 0.03 | 0.90 | 0.37 | 2.5 | 3.8 |
| GPT-4.1 | 0.10 | 1.4 | 0.57 | 1.7 | 3.8 |
| GPT-4o | 0.10 | 0.66 | 0.71 | 2.2 | 3.7 |
| Gemini-1.5-flash | 0.06 | 0.75 | 0.38 | 2.4 | 3.6 |
| Qwen2.5-VL-72B | 0.07 | 1.2 | 0.51 | 1.7 | 3.5 |
| Pixtral-12b | 0.00 | 0.81 | 0.00 | 2.6 | 3.4 |
| Nova-Pro | 0.08 | 0.62 | 0.68 | 1.7 | 3.0 |
| Llama-3.2-90B | 0.09 | 0.08 | 0.20 | 2.7 | 3.0 |
| GPT-4o-mini | 0.07 | 0.43 | 0.41 | 2.0 | 3.0 |
| GPT-4-turbo | 0.10 | 0.54 | 0.94 | 1.1 | 2.7 |
| Grok-2-Vision | 0.10 | 0.38 | 1.0 | 1.0 | 2.5 |
| Llama-3.2-11B | 0.01 | 0.65 | 0.00 | 1.6 | 2.3 |
| Qwen2.5-VL-7B | 0.03 | 1.0 | 0.32 | 0.79 | 2.2 |
| Nova-Lite | 0.10 | 0.22 | 0.24 | 1.6 | 2.1 |
| GPT-4.1-nano | 0.10 | 0.13 | 0.19 | 1.4 | 1.9 |
| Claude-3-haiku | 0.10 | 0.00 | 0.63 | 0.09 | 0.82 |
| Claude-3-sonnet | 0.10 | 0.07 | 0.50 | 0.00 | 0.67 |
| MiniMax-01 | 0.00 | 0.00 | 0.00 | 0.00 | 0.00 |

Table 4: Estimated $\theta$ on SEEDBENCH

| model | $\theta_{\text{base}}$ | $\theta_{\text{image}}$ | $\theta_{\text{text}}$ | $\theta_{\text{cross}}$ | total |
|---|---|---|---|---|---|
| GPT-4.1 | 0.10 | 2.0 | 1.9 | 2.0 | 6.0 |
| GPT-4o | 0.00 | 2.0 | 0.51 | 2.0 | 4.5 |
| GPT-4.1-mini | 0.10 | 2.0 | 2.0 | 0.00 | 4.1 |
| Gemini-2.0-flash | 0.10 | 2.0 | 2.0 | 0.00 | 4.1 |
| Gemini-1.5-pro | 0.10 | 2.0 | 2.0 | 0.00 | 4.1 |
| Qwen-2.5-VL-72b | 0.10 | 2.0 | 2.0 | 0.00 | 4.1 |
| Gemini-1.5-flash | 0.10 | 1.7 | 2.0 | 0.00 | 3.8 |
| Claude-3.5-sonnet | 0.00 | 0.80 | 0.00 | 2.0 | 2.8 |
| GPT-4o-mini | 0.00 | 2.0 | 0.58 | 0.07 | 2.6 |
| GPT-4.1-nano | 0.00 | 0.98 | 1.6 | 0.00 | 2.6 |
| Claude-3.7-sonnet | 0.10 | 0.35 | 0.00 | 2.0 | 2.4 |
| Nova-Lite | 0.00 | 1.9 | 0.00 | 0.54 | 2.4 |
| MiniMax-01 | 0.00 | 2.0 | 0.39 | 0.00 | 2.4 |
| Grok-2-Vision | 0.10 | 1.4 | 0.23 | 0.43 | 2.2 |
| Pixtral-Large | 0.00 | 1.3 | 0.45 | 0.02 | 1.8 |
| Pixtral-12b | 0.00 | 1.2 | 0.30 | 0.00 | 1.5 |
| Nova-Pro | 0.00 | 1.1 | 0.00 | 0.00 | 1.1 |
| Gemini-1.5-flash-8b | 0.00 | 0.96 | 0.00 | 0.00 | 0.96 |
| Qwen-2.5-VL-7b | 0.00 | 0.63 | 0.09 | 0.00 | 0.73 |
| GPT-4-turbo | 0.00 | 0.00 | 0.00 | 0.00 | 0.00 |
| Claude-3-haiku | 0.00 | 0.00 | 0.00 | 0.00 | 0.00 |

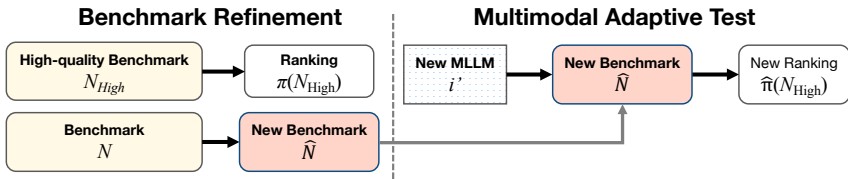

Figure 15: An illustration of our benchmark refinement and testing processes.

## D    OMITTED RESULTS OF MULTIMODAL BENCHMARK REFINEMENT

First, we illustrate our problem setting for benchmark refinement in Fig. 15. To investigate how the estimated parameters of the original questions and low-quality questions vary, we show the distribution of the estimated difficulty, discrimination, and the Fisher information of the original questions and the low-quality questions in Fig. 16 and Fig. 17. We also investigated whether there is a significant difference between the two distributions with the Mann-Whitney U test. Asterisks mark

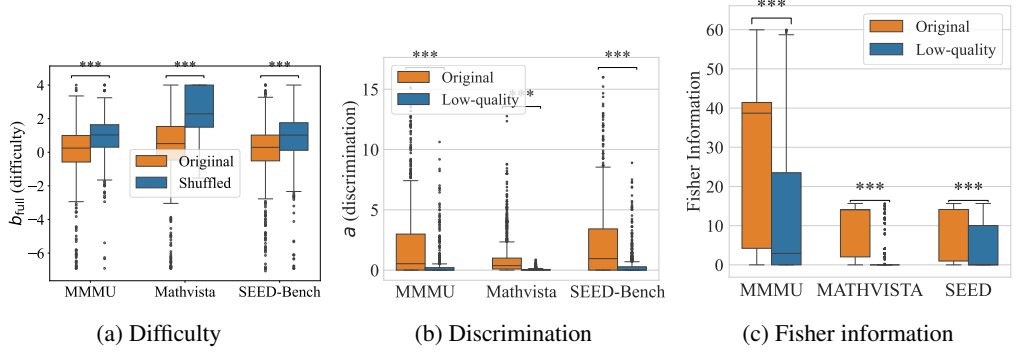

(a) Difficulty            (b) Discrimination            (c) Fisher information

Figure 16: Comparisons of parameters estimated by M2IRT between the original and artificial questions.

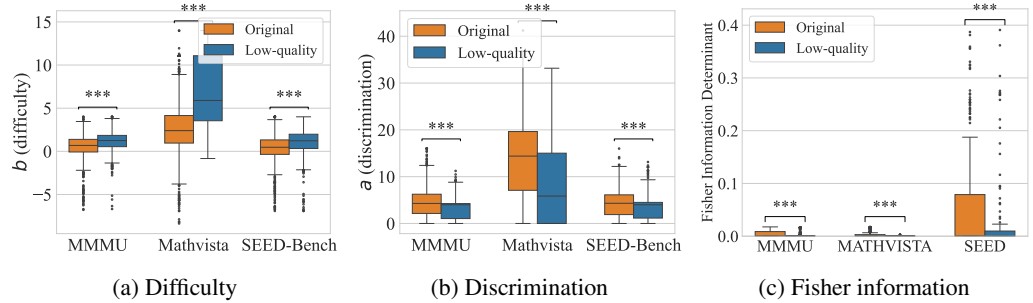

Figure 17: Comparisons of parameters estimated by M3IRT between the original and artificial questions.

the Mann-Whitney U test results comparing the original questions with the low-quality questions. [2] We confirmed significant differences between the groups of original and low-quality questions.

We additionally examined the Wasserstein distance between original and low-quality questions, and found that the Wasserstein distance for MMMU, MATHVISTA, and SEED-BENCH were 0.20, 0.14, and 0.051, respectively.

## D.1 DETAILED RESULTS

Detailed results of the experiments depicted in Fig. 4 are reported in Table 5. Detailed results of the experiments depicted in Fig. 5 are reported in Table 6.

Table 5: The average and standard deviation of Spearman's rank correlations between model rankings on the original benchmark and those estimated on extracted question subsets with 10%, 20%, 30%, 40%, and 50% of whole dataset.

| Benchmark | Method | 10% | 20% | 30% | 40% | 50% |
|---|---|---|---|---|---|---|
| MMMU | M2IRT | $0.96 \pm 0.061$ | $0.96 \pm 0.057$ | $0.96 \pm 0.047$ | $0.97 \pm 0.042$ | $0.96 \pm 0.035$ |
| | M3IRT | $0.89 \pm 0.046$ | $0.92 \pm 0.030$ | $0.92 \pm 0.033$ | $0.93 \pm 0.023$ | $0.93 \pm 0.023$ |
| | IRT | $0.50 \pm 0.21$ | $0.36 \pm 0.25$ | $0.38 \pm 0.20$ | $0.35 \pm 0.23$ | $0.12 \pm 0.28$ |
| | MIRT | $0.16 \pm 0.089$ | $0.33 \pm 0.091$ | $0.47 \pm 0.11$ | $0.61 \pm 0.076$ | $0.69 \pm 0.055$ |
| | TinyBenchmarks | $0.43 \pm 0.13$ | $0.50 \pm 0.12$ | $0.56 \pm 0.11$ | $0.61 \pm 0.11$ | $0.67 \pm 0.082$ |
| | FlashEval | $0.79 \pm 0.029$ | $0.79 \pm 0.025$ | $0.79 \pm 0.024$ | $0.80 \pm 0.017$ | $0.80 \pm 0.015$ |
| | Random | $0.77 \pm 0.062$ | $0.80 \pm 0.027$ | $0.82 \pm 0.028$ | $0.82 \pm 0.029$ | $0.81 \pm 0.024$ |
| MathVista | M2IRT | $0.81 \pm 0.036$ | $0.84 \pm 0.032$ | $0.88 \pm 0.042$ | $0.91 \pm 0.034$ | $0.93 \pm 0.017$ |
| | M3IRT | $0.92 \pm 0.028$ | $0.94 \pm 0.018$ | $0.93 \pm 0.022$ | $0.93 \pm 0.018$ | $0.93 \pm 0.010$ |
| | IRT | $0.81 \pm 0.047$ | $0.88 \pm 0.040$ | $0.91 \pm 0.023$ | $0.93 \pm 0.017$ | $0.94 \pm 0.014$ |
| | MIRT | $0.58 \pm 0.049$ | $0.67 \pm 0.038$ | $0.72 \pm 0.029$ | $0.76 \pm 0.026$ | $0.81 \pm 0.022$ |
| | TinyBenchmarks | $0.79 \pm 0.038$ | $0.84 \pm 0.020$ | $0.86 \pm 0.018$ | $0.88 \pm 0.017$ | $0.88 \pm 0.011$ |
| | FlashEval | $0.89 \pm 0.020$ | $0.91 \pm 0.015$ | $0.91 \pm 0.010$ | $0.91 \pm 0.014$ | $0.91 \pm 0.008$ |
| | Random | $0.89 \pm 0.040$ | $0.91 \pm 0.029$ | $0.92 \pm 0.018$ | $0.93 \pm 0.015$ | $0.93 \pm 0.013$ |
| SEED-Bench | M2IRT | $0.94 \pm 0.008$ | $0.96 \pm 0.006$ | $0.97 \pm 0.0080$ | $0.97 \pm 0.005$ | $0.95 \pm 0.005$ |
| | M3IRT | $0.94 \pm 0.019$ | $0.95 \pm 0.013$ | $0.95 \pm 0.011$ | $0.95 \pm 0.018$ | $0.95 \pm 0.017$ |
| | IRT | $0.86 \pm 0.13$ | $0.83 \pm 0.18$ | $0.84 \pm 0.15$ | $0.81 \pm 0.18$ | $0.79 \pm 0.17$ |
| | MIRT | $0.42 \pm 0.120$ | $0.59 \pm 0.074$ | $0.71 \pm 0.052$ | $0.77 \pm 0.045$ | $0.82 \pm 0.035$ |
| | TinyBenchmarks | $0.69 \pm 0.057$ | $0.75 \pm 0.050$ | $0.77 \pm 0.056$ | $0.80 \pm 0.049$ | $0.81 \pm 0.044$ |
| | FlashEval | $0.89 \pm 0.021$ | $0.90 \pm 0.015$ | $0.90 \pm 0.013$ | $0.90 \pm 0.010$ | $0.90 \pm 0.009$ |
| | Random | $0.86 \pm 0.046$ | $0.87 \pm 0.032$ | $0.89 \pm 0.027$ | $0.91 \pm 0.025$ | $0.91 \pm 0.011$ |

## D.2 VQAAT

**VQA-ANSWERTHERAPY** (VQAAT) (Chen et al., 2023) consists of VizWiz Dataset (Gurari et al., 2018), which is visual questions asked by visually impaired people, and VQA v2.0 (Goyal et al., 2017). We randomly sample 1000 questions from the train and validation sets of Single Answer

---

[2]Significance follows: $^*p < 0.05$, $^{**}p < 0.01$, $^{***}p < 0.001$.

Table 6: The average and standard deviation of the proportions of the low-quality questions in extracted question subsets $\gamma$ with 10%, 20%, 30%, 40%, and 50% of whole dataset. "TB" means TinyBenchmarks, and "FE" means FlashEval.

| Benchmark | Method | 10% | 20% | 30% | 40% | 50% |
|---|---|---|---|---|---|---|
| MMMU | M2IRT | $0.038 \pm 0.015$ | $0.090 \pm 0.014$ | $0.12 \pm 0.014$ | $0.16 \pm 0.014$ | $0.19 \pm 0.013$ |
| | M3IRT | $0.091 \pm 0.017$ | $0.11 \pm 0.013$ | $0.14 \pm 0.012$ | $0.16 \pm 0.011$ | $0.19 \pm 0.0070$ |
| | IRT | $0.32 \pm 0.040$ | $0.34 \pm 0.035$ | $0.36 \pm 0.029$ | $0.37 \pm 0.023$ | $0.38 \pm 0.020$ |
| | MIRT | $0.42 \pm 0.020$ | $0.40 \pm 0.016$ | $0.38 \pm 0.012$ | $0.37 \pm 0.012$ | $0.36 \pm 0.011$ |
| | TB | $0.38 \pm 0.023$ | $0.39 \pm 0.021$ | $0.38 \pm 0.022$ | $0.37 \pm 0.021$ | $0.36 \pm 0.020$ |
| | FE | $0.28 \pm 0.014$ | $0.29 \pm 0.017$ | $0.29 \pm 0.021$ | $0.27 \pm 0.017$ | $0.29 \pm 0.010$ |
| | Random | $0.31 \pm 0.031$ | $0.30 \pm 0.015$ | $0.30 \pm 0.010$ | $0.30 \pm 0.0081$ | $0.30 \pm 0.0080$ |
| MathVista | M2IRT | $0.011 \pm 0.0060$ | $0.037 \pm 0.014$ | $0.068 \pm 0.021$ | $0.11 \pm 0.021$ | $0.15 \pm 0.016$ |
| | M3IRT | $0.054 \pm 0.013$ | $0.11 \pm 0.014$ | $0.15 \pm 0.019$ | $0.17 \pm 0.013$ | $0.20 \pm 0.0078$ |
| | IRT | $0.27 \pm 0.055$ | $0.25 \pm 0.037$ | $0.24 \pm 0.028$ | $0.23 \pm 0.020$ | $0.22 \pm 0.018$ |
| | MIRT | $0.50 \pm 0.024$ | $0.43 \pm 0.023$ | $0.39 \pm 0.018$ | $0.36 \pm 0.013$ | $0.33 \pm 0.0083$ |
| | TB | $0.33 \pm 0.027$ | $0.31 \pm 0.019$ | $0.30 \pm 0.015$ | $0.29 \pm 0.010$ | $0.28 \pm 0.0065$ |
| | FE | $0.27 \pm 0.016$ | $0.28 \pm 0.018$ | $0.30 \pm 0.016$ | $0.31 \pm 0.015$ | $0.30 \pm 0.014$ |
| | Random | $0.29 \pm 0.031$ | $0.29 \pm 0.017$ | $0.29 \pm 0.0086$ | $0.29 \pm 0.0074$ | $0.29 \pm 0.0070$ |
| SEED-Bench | M2IRT | $0.045 \pm 0.016$ | $0.086 \pm 0.011$ | $0.13 \pm 0.0074$ | $0.17 \pm 0.0062$ | $0.22 \pm 0.0037$ |
| | M3IRT | $0.14 \pm 0.018$ | $0.14 \pm 0.012$ | $0.18 \pm 0.011$ | $0.21 \pm 0.0091$ | $0.24 \pm 0.0080$ |
| | IRT | $0.31 \pm 0.048$ | $0.33 \pm 0.041$ | $0.34 \pm 0.040$ | $0.35 \pm 0.038$ | $0.37 \pm 0.031$ |
| | MIRT | $0.43 \pm 0.023$ | $0.41 \pm 0.013$ | $0.39 \pm 0.012$ | $0.38 \pm 0.0083$ | $0.37 \pm 0.0087$ |
| | TB | $0.36 \pm 0.015$ | $0.35 \pm 0.013$ | $0.35 \pm 0.012$ | $0.34 \pm 0.0082$ | $0.34 \pm 0.0081$ |
| | FE | $0.30 \pm 0.019$ | $0.31 \pm 0.020$ | $0.33 \pm 0.014$ | $0.34 \pm 0.015$ | $0.34 \pm 0.016$ |
| | Random | $0.34 \pm 0.036$ | $0.34 \pm 0.021$ | $0.34 \pm 0.011$ | $0.34 \pm 0.0084$ | $0.34 \pm 0.0084$ |

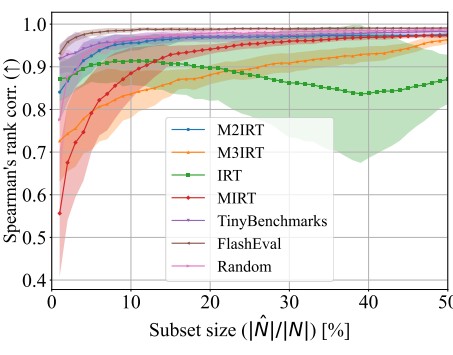

(a) Spearman's rank correlations on VQAAT

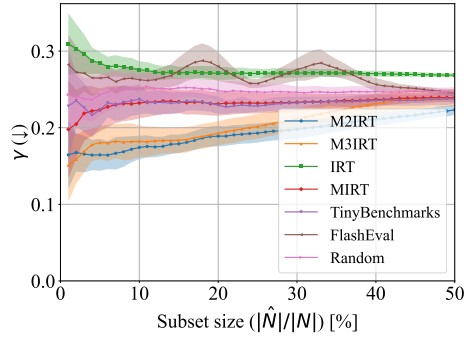

(b) The proportions of the low-quality questions $\gamma$ on VQAAT

Figure 18: The average and standard deviation of Spearman's rank correlations on extracted question subsets of VQAAT and the proportions of the low-quality questions $\gamma$ in extracted question subsets with different sizes.

Grounding Challenge. This dataset presents images, questions, and multiple annotators' responses to those questions to the VLM, asking whether the annotators' answers are based on the same part of the image. Therefore, this dataset consists solely of binary-choice questions.

We conducted an additional experiment on VQAAT under the same condition as Section 5.3. Figure 4 shows the Spearman's rank correlations between the model rankings on the original benchmark and on an extracted subset and proportions of the low-quality questions. In contrast to experiment in Section 5.3, on VQAAT, M2IRT and M3IRT are worse than baselines in terms of the Spearman's rank correlation. Interestingly, while proposed methods extracts fewer low-quality questions than the baselines, this filtering does not translate to improved accuracy in model ranking. We hypothesize that the discrepancy arises because VQAAT itself contains numerous low-quality questions, significantly influencing its "ground truth" ranking. As our method filters such low-quality questions, the resulting ranking deviates from the original benchmark.

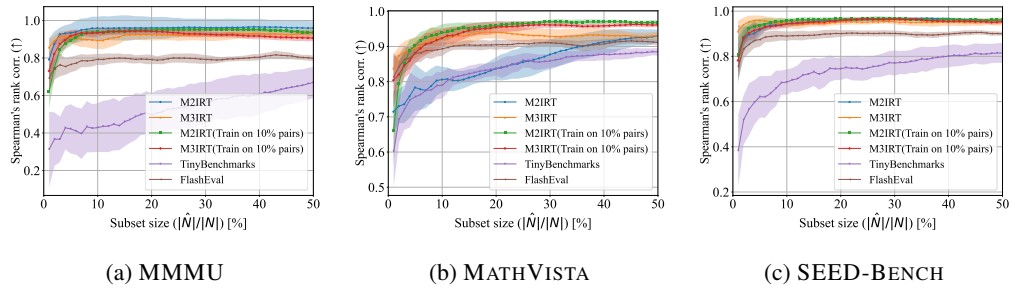

(a) MMMU        (b) MATHVISTA        (c) SEED-BENCH

Figure 19: The average and standard deviation of Spearman's rank correlations between model rankings on the original benchmark and those estimated on extracted question subsets with different sizes.M2IRT and M3IRT are trained with sparse-response matrix.

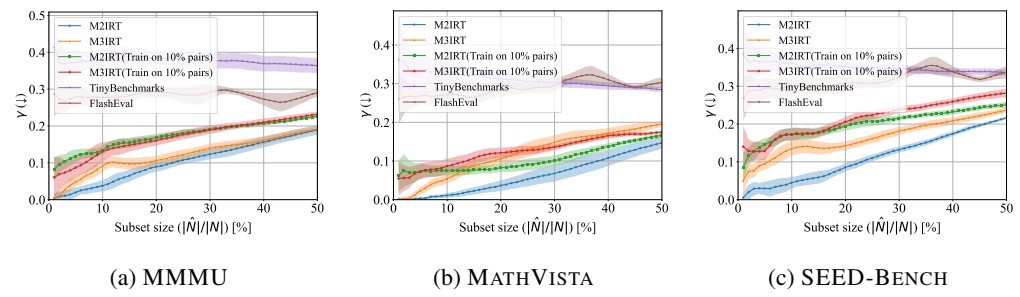

(a) MMMU        (b) MATHVISTA        (c) SEED-BENCH

Figure 20: The average and standard deviation of the proportions of the low-quality questions in extracted question subsets $\gamma$ with different sizes. M2IRT and M3IRT are trained with sparse-response matrix.

## D.3 FOR SPARSE RESPONSE MATRIX

In Section 4.4, we explain that M2IRT and M3IRT don't require all models to respond to all questions. To demonstrate this, we conducted an additional experiment identical to Section 5.3, except that we used only 10% of all (model, question) pairs on MMMU for training M2IRT and M3IRT. We then selected informative questions for evaluating a new model. These models trained with sparse dataset are compared with original and strong baselines.

Figure 19 shows the Spearman's rank correlations between the model rankings on the original benchmark and on different sizes of subsets. Figure 20 shows the proportion $\gamma$ with varying size of subsets.

Remarkably, as shown in Fig. 19a using just 3% of the total questions, M3IRT trained only 10% of all (model, question) pairs achieved a Spearman rank correlation exceeding 0.84 with the original full-dataset rankings—equivalent to the baseline performance that requires 50% of the dataset. Beyond this point, M2IRT consistently maintained higher ranking consistency than all baselines. Moreover, the proportion of low-quality questions selected remained below 23%, showing that M2IRT is not only efficient but also discriminative in identifying high-quality items.

This setting allows model comparison at only 13% of the inference cost required for full evaluation across all models and questions, demonstrating that M2IRT offers substantial cost savings while preserving evaluation reliability.

## D.4 STATISTICAL SIGNIFICANCE TESTS

We conducted a one-sided Wilcoxon signed-rank test to evaluate the performance difference between M2IRT and FlashEval,M2IRT and TinyBenchmarks, M3IRT and FlashEval,and M3IRT and Tiny-Benchmarks. Table 7, Table 8, and Table 9 show the results for 5%, 10%, 30%, and 50% of Fig. 4 and Fig. 5 with a confidence level of 1%. MMMU shows significant differences from the baseline

method. MATHVISTA shows significant differences in all conditions except for Spearman's rank corr at 5% against FlashEval's score. SEED-BENCH shows significant differences from the baseline method.

Table 7: Wilcoxon signed-rank test on MMMU comparing FlashEval and TinyBench against M3IRT.

| Comparison | 5% subset | | 10% subset | | 30% subset | | 50% subset | |
|---|---|---|---|---|---|---|---|---|
| | $p$-value | $W$ | $p$-value | $W$ | $p$-value | $W$ | $p$-value | $W$ |
| vs FlashEval (Rank corr.) | $< 0.0001$ | 0.0 | $< 0.0001$ | 0.0 | $< 0.0001$ | 0.0 | $< 0.0001$ | 0.0 |
| vs TinyBench (Rank corr.) | $< 0.0001$ | 0.0 | $< 0.0001$ | 0.0 | $< 0.0001$ | 0.0 | $< 0.0001$ | 0.0 |
| vs FlashEval (Shuffle ratio) | $< 0.0001$ | 0.0 | $< 0.0001$ | 0.0 | $< 0.0001$ | 0.0 | $< 0.0001$ | 0.0 |
| vs TinyBench (Shuffle ratio) | $< 0.0001$ | 0.0 | $< 0.0001$ | 0.0 | $< 0.0001$ | 0.0 | $< 0.0001$ | 0.0 |

Table 8: Wilcoxon signed-rank test on MATHVISTA comparing FlashEval and TinyBench against M3IRT.

| Comparison | 5% subset | | 10% subset | | 30% subset | | 50% subset | |
|---|---|---|---|---|---|---|---|---|
| | $p$-value | $W$ | $p$-value | $W$ | $p$-value | $W$ | $p$-value | $W$ |
| vs FlashEval(Rank Corr.) | 0.0197 | 222.0 | 0.0004 | 262.0 | 0.0019 | 251.5 | $< 0.0001$ | 293.0 |
| vs TinyBench(Rank Corr.) | $< 0.0001$ | 300.0 | $< 0.0001$ | 300.0 | $< 0.0001$ | 300.0 | $< 0.0001$ | 300.0 |
| vs FlashEval(Shuffle ratio) | $< 0.0001$ | 0.0 | $< 0.0001$ | 0.0 | $< 0.0001$ | 0.0 | $< 0.0001$ | 0.0 |
| vs TinyBench(Shuffle ratio) | $< 0.0001$ | 0.0 | $< 0.0001$ | 0.0 | $< 0.0001$ | 0.0 | $< 0.0001$ | 0.0 |

Table 9: Wilcoxon signed-rank test on SEEDBench comparing FlashEval and TinyBench against M3IRT.

| Comparison | 5% subset | | 10% subset | | 30% subset | | 50% subset | |
|---|---|---|---|---|---|---|---|---|
| | $p$-value | $W$ | $p$-value | $W$ | $p$-value | $W$ | $p$-value | $W$ |
| vs FlashEval (Rank corr.) | $< 0.0001$ | 231.0 | $< 0.0001$ | 231.0 | $< 0.0001$ | 231.0 | $< 0.0001$ | 231.0 |
| vs TinyBench (Rank corr.) | $< 0.0001$ | 231.0 | $< 0.0001$ | 231.0 | $< 0.0001$ | 231.0 | $< 0.0001$ | 231.0 |
| vs FlashEval (Shuffle ratio) | $< 0.0001$ | 0.0 | $< 0.0001$ | 0.0 | $< 0.0001$ | 0.0 | $< 0.0001$ | 0.0 |
| vs TinyBench (Shuffle ratio) | $< 0.0001$ | 0.0 | $< 0.0001$ | 0.0 | $< 0.0001$ | 0.0 | $< 0.0001$ | 0.0 |

# E DETAILS OF EXPERIMENTAL SETTINGS

## E.1 COMPUTATIONAL RESOURCES

The computational resources utilized in this study are presented in Table 10. The experiments in Section 5.3 require 2 hours per dataset, and those in Section 5.4 necessitate 3 hours per dataset.

## E.2 DATASETS

**A Massive Multi-discipline Multimodal Understanding and Reasoning Benchmark for Expert AGI (MMMU)** (Yue et al., 2024) : The license for this dataset is "Apache License 2.0".

**MATHVISTA** (Lu et al., 2024) : The license for this dataset is "Creative Commons Attribution Share Alike 4.0 International".

**VQA-ANSWERTHERAPY** (VQAAT) (Chen et al., 2023) : The license for this dataset is "Creative Commons Attribution 4.0 International License".

**SEED-BENCH** (Li et al., 2024a) : The license for this dataset is "Creative Commons Attribution Non Commercial 4.0".

## E.3 VLMS

We use 24 commonly used VLMs listed in Table 11 for our experiments. We access open-source models and a subset of closed models through Openrouter.

Table 10: Computer Specifications Used for Experiments

| Component | Specification |
|---|---|
| Operating System | Ubuntu 20.04 LTS |
| CPU | AMD EPYC Milan 7763 DP/UP (64C/128T, 2.45GHz) $\times$ 2 |
| Memory | 2048GB |
| python version | 3.12.9 |
| torch version | 2.6.0 |

Table 11: Overview of AI Models Used

| Model Name | Type | License or Terms |
|---|---|---|
| GPT-4-turbo | Closed | OpenAI Terms of Use |
| GPT-4o (OpenAI, 2024a) | Closed | OpenAI Terms of Use |
| GPT-4o-mini (OpenAI, 2024b) | Closed | OpenAI Terms of Use |
| GPT-4.1 (OpenAI, 2025) | Closed | OpenAI Terms of Use |
| GPT-4.1-mini (OpenAI, 2025) | Closed | OpenAI Terms of Use |
| GPT-4.1-nano (OpenAI, 2025) | Closed | OpenAI Terms of Use |
| Gemini-1.5-flash (Team, 2024) | Closed | Gemini API Additional Terms of Service |
| Gemini-1.5-flash-8b (Team, 2024) | Closed | Gemini API Additional Terms of Service |
| Gemini-1.5-pro (Team, 2024) | Closed | Gemini API Additional Terms of Service |
| Gemini-2.0-flash (Pichai, 2024) | Closed | Gemini API Additional Terms of Service |
| Claude-3-haiku (Anthropic, 2024a) | Closed | Anthropic Consumer Terms of Service |
| Claude-3-sonnet (Anthropic, 2024a) | Closed | Anthropic Consumer Terms of Service |
| Claude-3.5-sonnet (Anthropic, 2024b) | Closed | Anthropic Consumer Terms of Service |
| Claude-3.7-sonnet (Anthropic, 2025) | Closed | Anthropic Consumer Terms of Service |
| Grok-2 (xAI, 2024) | Closed | xAI Terms of Service |
| Nova-pro (Intelligence, 2024) | Closed | AWS Terms of Service |
| Nova-lite (Intelligence, 2024) | Closed | AWS Terms of Service |
| Qwen-2.5-vl-7b (Bai et al., 2025) | Open | Apache 2.0 |
| Qwen-2.5-vl-72b (Bai et al., 2025) | Open | Apache 2.0 |
| Llama-3.2-11b-instruct (Meta, 2024) | Open | Llama 3.2 Community License |
| Llama-3.2-90b-instruct (Meta, 2024) | Open | Llama 3.2 Community License |
| Pixtral-12b (Agrawal et al., 2024) | Open | Apache 2.0 |
| Pixtral-large (Agrawal et al., 2024) | Open | Apache 2.0 |
| Minimax-01 (Team, 2025) | Open | MIT License |

