# OpenReview forum: "Evaluating Cross-Modal Reasoning Ability and Problem Characteristics with Multimodal Item Response Theory"
_ICLR.cc/2026/Conference — ICLR 2026 Poster_

### Official Review · Reviewer_jMkP · 2025-10-23

**Soundness:** 4
**Presentation:** 4
**Contribution:** 4
**Rating:** 6
**Confidence:** 3

**Summary:**

This paper addresses a key deficiency in current benchmarks for Multimodal Large Language Models (MLLMs): the prevalence of "shortcut questions" that can be answered using only a single modality, such as text or image alone. This flaw leads to unreliable model rankings and unnecessarily increases evaluation costs due to large benchmark sizes. To resolve this, the authors propose a novel framework called Multimodal and Multidimensional Item Response Theory (M³-IRT). This approach extends classical Item Response Theory (IRT) by decomposing both the model's ability and the question's difficulty into three latent components: image-only, text-only, and cross-modal integration.

This decomposition allows the framework to precisely identify high-quality questions that genuinely require cross-modal reasoning while filtering out low-quality shortcuts. The method was validated through experiments on 24 Vision-Language Models (VLMs) across three benchmarks: MMMU, MATHVISTA, and SEED-BENCH. The results demonstrate that M³-IRT is robust against contamination, maintaining ranking fidelity even when benchmarks contain up to 50% artificially generated low-quality questions. Furthermore, the framework enables the creation of compact, high-quality benchmark subsets that can accurately reproduce the full ranking using only a small fraction of the questions, thereby significantly reducing computational overhead while improving evaluation reliability.

**Strengths:**

1. The paper's primary strength lies in its novel framework, M³-IRT, which moves beyond simple accuracy scores to offer a more granular and interpretable evaluation.

2. The framework enables the creation of compact, high-quality benchmark subsets that significantly reduce the computational cost of evaluation.

3. The paper effectively demonstrates the method's resilience to the low-quality and "shortcut" questions that are common in existing benchmarks.

**Weaknesses:**

1.  The decomposition of a model's ability, theta ($\theta$), is a heuristic and oversimplified mathematical model. It operates on the assumption that the distinct abilities (e.g., image, text, cross-modal) are independent and linearly additive. However, a model's actual reasoning ability is complex. For instance, a model might leverage textual concepts to guide its attention within an image, creating a synergistic effect where cross-modal ability mutually enhances the image ability. Such complex, non-linear interactions cannot be captured by the current linear additive framework.

2.  Given that numerous parameters are optimized via SGD, the stability of $\theta$ across benchmarks of varying scales has not been adequately validated. A critical question remains unanswered: if the number of questions were to substantially increase or decrease, would a model's ability score exhibit significant fluctuations? This lack of demonstrated robustness raises concerns about the reliability of the scores when the evaluation context changes.

3.  The model's ability ($\theta$), question difficulty ($b$), and discrimination ($a$) are all optimized simultaneously. Consequently, the difficulty of a question is inherently influenced by the abilities of the models in the test cohort. This renders all metrics as **relative scores** within the context of a specific group (24 VLMs in this paper), rather than absolute measures. If the composition or average capability of the VLM cohort were to change, the perceived difficulty of the questions would also shift, raising concerns about the potential for instability in the evaluation framework.

**Questions:**

1. The $\theta_{base}$ in Supp. Table 2: Mathvista shows that GPT-4o's base is 0.00 and Llama-3.2-11B is 0.10, which is weird. Can authors explain this phenomenon?

2. Given that the evaluation scores are relative to the tested models, how do you address the potential instability of these scores as the model evolves?

---

> ### Author Response · Authors · 2025-11-26
> **Response to Reviewer jMkP**
>
> We appreciate your comments on the structure and stability of our model, as well as the experimental results.
> ## > W1. The decomposition of a model's ability, theta, is a heuristic and oversimplified mathematical model.
>
> Our goal is not merely to improve prediction accuracy, but to visualize and interpret the contribution of each modality and the cross-modality component. Therefore, we adopt linear decomposition as the minimal structure that satisfies this requirement, which is precisely the IRT approach. While a nonlinear model could potentially improve accuracy, it would compromise the separation of contributions of modalities and the clarity of interpretation.
>
> We evaluated an additional nonlinear model that replaces the cross-modality term $b_{\text{cross}}$ with a cross term $b_{\text{image}} \times b_{\text{text}}$.  Please refer our comments to W3 of Reviewer akTA.
>
> ## > W2. if the number of questions were to substantially increase or decrease, would a model's ability score exhibit significant fluctuations?
>
> We evaluated how the model's ability score changes based on the number of questions used. We show RMSE between the ability score estimated using all questions and those estimated using 90%, 60%, 30%, 10%, and 1% of randomly selected questions in Table R5.
>
> We confirmed that the change is small for SEED-Bench and MathVista, but large for MMMU. This difference stems from the characteristics of the benchmarks, suggesting that SEED-Bench and MathVista are more homogeneous and are more robust to question reduction than MMMU. This larger RMASE in MMMU likely reflects its heterogeneous nature covering six disciplines.
>
> **Table R5: RMSE of estimated ability scores for varying proportions of questions**
> |Proportion of questions|MMMU|MathVista|SEED-Bench|
> |--|--|--|--|
> |90%|1.3 $\pm$ 1.8|1.4 $\pm$ 1.0|0.49 $\pm$ 0.20|
> |60%|4.0 $\pm$ 3.1|1.7 $\pm$ 0.93|0.58 $\pm$ 0.21|
> |30%|6.4 $\pm$ 2.4|2.1 $\pm$ 0.72|1.1 $\pm$ 0.32|
> |10%|8.1 $\pm$ 1.0|2.4 $\pm$ 0.49|1.2 $\pm$ 0.23|
> |1%|9.7 $\pm$ 1.1|2.8 $\pm$ 0.26|1.7 $\pm$ 0.12|
>
> ## > W3. the potential for instability in the evaluation framework.
>
> IRT provides a theoretical guarantee of parameter invariance, meaning the same parameters are estimated from any population. However, since our M^3-IRT is a variant of IRT and the number of VLM is limited to 24, we conducted additional verification and confirmed that the estimated parameters are stable when a sufficient number of VLMs are available.
>
> IRT defines item parameters independently of examinee ability [d]. Thus, even when using different VLMs, the same item parameters should be estimated given a large number of VLMs. When VLM cohorts differ significantly, we can calibrate estimation results using Equating or Linking [e], which is a method long employed in standardized tests like the GMAT and GRE.
>
> Based on your comment, we evaluated how the number of VLM changes the estimated parameters. We show RMSE between the difficulty parameters estimated from all 24 VLM and those from 75% or 50% of randomly selected VLMs in Table R6. While RMSE tends to increase as the number of VLM decreases, it remains relatively small at 75%. We believe using a larger number of VLM would yield more stable results in the future.
>
> [d] Lord, F. M. (1980). Applications of Item Response Theory to Practical Testing Problems.
> [e] Kolen, M. J., & Brennan, R. L. (2014). Test Equating, Scaling, and Linking: Methods and Practices (3rd ed.).
>
> **Table R6:  RMSE of estimated difficulties for varying proportions of LLMs**
> ||MMMU|MathVista|SEED-Bench|
> |--|--|--|--|
> |75%|1.4 $\pm$ 0.32|0.81 $\pm$ 0.059|0.66 $\pm$ 0.010|
> |50%|2.0 $\pm$ 0.35|1.2 $\pm$ 0.15|0.9 $\pm$ 0.13|
>
> ## > Q1 The  in Supp. Table 2: Mathvista shows that GPT-4o's base is 0.00 and Llama-3.2-11B is 0.10, which is weird. Can authors explain this phenomenon?
>
> We investigated the accuracy of GPT-4o and Llama-3.2-11B when only options were presented in MathVista and showed the result in Table R7. We confirmed that GPT-4o (0.059) was less accurate than Llama-3.2-11B (0.172). We believe that M^3-IRT appropriately estimated the base ability. We will include a new table showing the accuracies for each format.
>
> **Table R7: the accuracy of GPT-4o and Llama-3.2-11B**
> || Option | Image+Option |Text+Option|Image+Text+Option|
> |--|--|--|--|--|
> | GPT-4o| 0.059 | 0.15 | 0.26 | 0.56 |
> | Llama-3.2-11B | 0.17 | 0.18 | 0.24 | 0.33 |
>
> ## > Q2 Given that the evaluation scores are relative to the tested models, how do you address the potential instability of these scores as the model evolves?
>
> When a high-performance VLM emerges, we can assese its capabilities by adding new difficult questions. This is similar to how standardized tests add difficult questions to their question pool in response to increases in examinee ability. By having examinees answer both old and new questions, we can estimate the characteristics of the new questions. Note that humans must be able to construct questions that VLMs cannot solve.

---

### Official Review · Reviewer_Wauf · 2025-10-23

**Soundness:** 3
**Presentation:** 2
**Contribution:** 2
**Rating:** 6
**Confidence:** 4

**Summary:**

This paper discusses the assessment of Multimodal Large Language Models (MLLMs). The authors observe that some multimodal problems in conventional datasets or benchmarks can be solved using only one modality. This makes current evaluation efforts of Multimodal Large Language Models less efficient and less reliable. The authors then propose their method, M3-IRT, which estimates the cross‑modal ability of MLLMs and each question’s cross‑modal difficulty, enabling compact, high‑quality subsets that better reflect multimodal reasoning.

**Strengths:**

1. This paper studies a very important problem in Multimodal Large Language Models, and the authors' observations about the weaknesses of existing evaluation efforts are reasonable.
2. This paper provides detailed descriptions of their evaluation framework. There are a lot of figures to visualize the results of evaluation.

**Weaknesses:**

1. The problem that this paper has pointed out (prior evaluation efforts use multimodal problems that can be solved with only one of the modalities) has been studied previously. This paper did not provide proper reference to these prior works. For example, MMEvalPro [1] is a recently proposed dataset with manually labeled questions to mitigate the problem.
2. The results are mostly displayed in the figures, while I expect more accurate numbers to be displayed in tables. While figures are effective for visualizing high-level trends and making qualitative comparisons, they are insufficient for a rigorous and reproducible scientific paper. It is difficult, and in some cases impossible, for the reader to extract the precise performance metrics, which is essential for understanding the exact magnitude of the reported improvements and for future comparative analysis by other researchers.
3. This paper should include a more comprehensive discussion of related works in MLLM evaluation, including [2-3].

[1] MMEvalPro: Calibrating Multimodal Benchmarks Towards Trustworthy and Efficient Evaluation
[2] Multifaceted Evaluation of Audio-Visual Capability for MLLMs: Effectiveness, Efficiency, Generalizability and Robustness
[3] Revisiting multi-modal llm evaluation

**Questions:**

Please refer to the weakness section.

---

> ### Author Response · Authors · 2025-11-26
> **Response to Reviewer Wauf**
>
> We are pleased to have the opportunity to discuss new related works and draw insights from them. We will include these works in our manuscript.
>
> ## > W1. MMEvalPro [1] is a recently proposed dataset with manually labeled questions to mitigate the problem.
>
> Thank you for suggesting the related research. MMEvalPro [R1] focuses on that MLLMs can answer questions using only textual knowledge without even looking at the image and achieve correct answers based on knowledge learnt from textbooks or existing QAs. However, their approach differs from ours because they aimed to expand and redesign benchmarks. Specifically, they employed annotators to add questions to check whether MLLMs properly recognize figures or images and whether MLLMs have the knowledge to solve questions. We will add this paper to our references. While MMEvalPro requires manual annotation to expand benchmarks, M^3-IRT provides an automated, post-hoc method without additional annotation costs.
> ## > W2. The results are mostly displayed in the figures, while I expect more accurate numbers to be displayed in tables.
>
> Thank you for your comments regarding the clarification of results. Due to space constraints, we presented the numerical evaluation results corresponding to Fig. 4 and Fig. 5 in Tables 4, 5, 6, 7, 8, and 9 of Appendix C.1. We agree that precise numerical values are crucial for ensuring the reliability and reproducibility of the research. We will either include the tables within the main text or reference them within the text. Additionally, we will create a table showing the numerical results for Fig. 6 and include it in the Appendix.
>
> ## > W3. This paper should include a more comprehensive discussion of related works in MLLM evaluation, including [2-3].
>
> Our research and [2] are related in evaluating the multimodal reasoning capabilities of MLLMs, though they focus on different modalities. The authors report that the performance of MLLMs handling Audio and Visual degrades when noise is added to the video or when only audio is input. Our work and [3] share a common motivation in measuring MLLM capabilities by focusing on benchmark shortcuts. However, their methodology differs from ours, as they are limited to task-based evaluation using multiple benchmarks.
>
> [1] Jinsheng Huang et al., 2025. "MMEvalPro: Calibrating Multimodal Benchmarks Towards Trustworthy and Efficient Evaluation". In Proceedings of the 2025 Conference of the Nations of the Americas Chapter of the Association for Computational Linguistics (NAACL).
>
> [2] Yusheng Zhao et al., 2025. “Multifaceted Evaluation of Audio-Visual Capability for MLLMs: Effectiveness, Efficiency, Generalizability and Robustness”. In Findings of the Association for Computational Linguistics (EMNLP).
>
> [3] Jian Lu et al., 2025. "Revisiting Multi-Modal LLM Evaluation". 2025 IEEE/CVF Conference on Computer Vision and Pattern Recognition Workshops (CVPRW).

---

### Official Review · Reviewer_akTA · 2025-10-28

**Soundness:** 3
**Presentation:** 2
**Contribution:** 2
**Rating:** 4
**Confidence:** 3

**Summary:**

To address unreliable rankings and high costs in MLLM evaluation caused by shortcut questions answerable from a single modality, this paper introduces the M³-IRT and M²-IRT frameworks. These extend classical Item Response Theory (IRT) by decomposing both model abilities and question difficulties into image-only, text-only, and cross-modal components. This allows for quantifying a model's cross-modal reasoning and an item's cross-modal demand. Experiments show the framework effectively prioritizes genuine cross-modal questions, faithfully reproducing full-benchmark rankings with as little as a 10% subset. It remains robust even when 50% of the benchmark is contaminated with low-quality items, significantly reducing evaluation costs while enhancing reliability.

**Strengths:**

1. The paper makes a contribution by addressing the critical challenge of "shortcut questions" in multimodal benchmarks. It provides a systematic solution that enhances the reliability of evaluations while simultaneously reducing computational costs.
2. The paper effectively targets two major pain points in current multimodal evaluation: unreliable model rankings caused by shortcut question contamination, and the high computational cost associated with large-scale benchmarks. The proposed M³-IRT framework offers an efficient and effective solution to both issues.
3. By decomposing model abilities into image-only, text-only, and cross-modal components, it offers valuable and interpretable insights into the specific strengths and weaknesses of different MLLMs, moving beyond a single, monolithic accuracy score.

**Weaknesses:**

1. The model's core assumption of linear decomposition for abilities and difficulties might oversimplify the complex, potentially non-linear interactions that occur during cross-modal reasoning.
2. The interpretability of the estimated parameters such as cross-modal difficulty is derived purely from model performance patterns and lacks external validation against human cognitive judgments of what constitutes a cross-modal task.
3. It is noted that the paper generates artificial low-quality questions through full swapping of images or text, which effectively simulates extreme scenarios involving complete modal mismatch. That said, it may be worth considering whether this approach fully covers the more prevalent and subtle shortcut features commonly observed in real-world multimodal benchmarks. In practical contexts, shortcuts often exhibit characteristics of concealment and diversity. Given that the framework’s ability to filter such subtle shortcuts has not yet been evaluated, it might be valuable to further explore its generalizability when applied to real-world benchmarks.

**Questions:**

Have you tested any non-linear variants? If not, how sensitive are your conclusions to the linearity assumption?

In addition, beyond swapping-based corruption, does your filter still identify low-quality items under subtler artifacts?

---

> ### Author Response · Authors · 2025-11-26
> **Response to Reviewer akTA**
>
> We appreciate your insightful comments and address the points you raised. We will include the discussion and tables in our manuscript.
>
> ## > W1. The model's core assumption of linear decomposition for abilities.
>
> Our goal is not merely to improve prediction accuracy, but to visualize and interpret the contribution of each modality and the cross-modality component. Therefore, we adopt linear decomposition as the minimal structure that satisfies this requirement, which is precisely the IRT approach. While a nonlinear model could potentially improve accuracy, it would compromise the separation of contributions of modalities and the clarity of interpretation, making it unsuitable for the objectives of this study.
>
> Based on your comment, to investigate the validity of linear decomposition, we evaluated an additional nonlinear model that replaces the cross-modality term $b_{\text{cross}}$ with a cross term $b_{\text{image}} \times b_{\text{text}}$.
>
> We show the rank correlations in Table R3. Both the linear and nonlinear models resulted in similar scores across all benchmarks. While the nonlinear model shows slightly higher rank correlations in some settings, the difference is small. We consider that our findings are not sensitive to the linearity assumption, and remain unchanged regardless of whether the cross-modal term is linear or nonlinear.
>
> **Table R3: Comparison of Rank Correlation for Linear and Nonlinear Models**
> |||5%|10%|30%|50%|
> |--|--|--|--|--|--|
> |MMMU| M^3-IRT|0.91|0.89| 0.92|0.93 |
> |MMMU| M^3-IRT w/ Non-linear|0.94|0.93|0.93|0.91|
> |MathVista| M^3-IRT| 0.90|0.92|0.93|0.93|
> |MathVista| M^3-IRT w/ Non-linear| 0.88|0.92|0.96|0.96|
> |SEED-Bench| M^3-IRT| 0.96|0.94|0.95|0.95|
> |SEED-Bench| M^3-IRT w/ Non-linear| 0.92|0.94|0.96|0.95|
> ## > W2. external validation against human cognitive judgments of what constitutes a cross-modal task.
>
> The parameters estimated by M^3-IRT represent the difficulty and modality requirements for VLMs, which may differ from those for humans. Based on your comment, we conducted an additional experiment with crowdsourcing to investigate the relationship between the degree of decline in human accuracy due to missing image or text and the cross-modal difficulty parameter $b_{\text{cross}}$. Our empirical findings can be summarized as follows: SEED-Bench showed weak correlation (0.35) with moderate inter-annotator agreement ($\kappa=0.44$). MMMU and MathVista showed almost no correlations, likely due to low agreement ($\kappa=0.25, 0.28$) or task difficulty for non-experts (Acc_{image+text}= 0.41, 0.23). The agreements among crowd workers are measured by Fleiss' kappa (where values above 0.6 are considered favorable) [a, b].
>
> **Detailed Experimental Design**: We randomly selected 200 questions per benchmark. Nine crowd workers answered each question under three formats (Image+Text, Image-only, Text-only). We calcurated the accuracy for each benchmark for each format. Then, we calcurated decline = Acc_{image+text}-(Acc_{image}+Acc_{text}}/2 and its Spearman's correlation with $b_{\text{cross}​}$.
>
> We show numerical results in Table R4. We consider that humans are not sufficiently capable of cross-modal reasoning on MMMU, as the decline is small. Furthermore, MathVista has a low accuracy rate for cloud workers, suggesting that evaluation is difficult for the general public and requires experts.
>
> **Table R4: Relationships between VLMs and human answer patterns**
> |  | MMMU  | MathVista | SEED-Bench |
> | -- | ----- | --------- | ---------- |
> | decline | 0.032 | -0.23 | 0.18 |
> | Fleiss' $\kappa$ | 0.25  | 0.28 | 0.44       |
> | Correlation between decline and $b_{\text{cross}}$ | 0.09  | -0.15 | 0.35 |
> | Acc_{image+text} of Crowd workers | 0.41| 0.23 | 0.62 |
> | Acc_{image+text} of VLMs  | 0.52  | 0.52| 0.57|
>
> [a] Fleiss (1971), Psychological Bulletin. [b] Gwet (2008), British J. Math. Stat. Psych.
> ## > W3. whether this approach fully covers the more prevalent and subtle shortcut features commonly observed in real-world multimodal benchmarks
>
> We considered a method for synthesizing realistic and complex low-quality datasets with prevalent and subtle shortcut features. For example, we can modify text and choices using crowdsourcing and LLMs, as well as adding noise to images. However, this editing method has the drawback that controlling the editing intensity is difficult, leading to confounding results. In contrast, simple swapping offers the advantage of enabling uniform and controlled editing.
>
> Since the experimental setup became extremely complex, we did not conduct additional experiments with subtler artifacts, but our qualitative analysis in Fig. 1 and Appendix B already demonstrates that M^3-IRT identifies naturally occurring subtle shortcuts in existing benchmarks.
>
> ## > Q1. Have you tested any non-linear variants?
>
> Please see our response to W1.
>
> ## > Q2. does your filter still identify low-quality items under subtler artifacts?
>
> Please see our response to W3.

---

> > ### Comment · Reviewer_akTA · 2025-11-26
> >
> > I appreciate the authors’ efforts and the additional clarifications provided in the rebuttal. However, I would like to maintain my original evaluation, which remains slightly below the acceptance threshold.

---

> > > ### Author Response · Authors · 2025-11-27
> > > **Response to Reviewer akTA**
> > >
> > > Thank you for your thoughtful review and for considering our rebuttal. To help us improve the work, could you clarify which specific aspects still keep the paper slightly below the acceptance threshold? Any brief guidance on what you feel remains insufficient would be greatly appreciated.

---

### Official Review · Reviewer_5cL5 · 2025-11-01

**Soundness:** 3
**Presentation:** 3
**Contribution:** 3
**Rating:** 6
**Confidence:** 4

**Summary:**

In this paper, the authors proposed M^3-IRT, a multi-modal and multidimensional item response theory framework to evaluate cross-modal reasoning ability of Multimodal Large Models (MLLMs). Multiple experiments were conducted with several VLMs to show that M^3-IRT could reduce evaluation cost while improving reliability.

**Strengths:**

1. Both the motivation of decomposing model ability and item difficulty and the proposed M^3-IRT framework make sense and are technically sound to me. Even though it builds upon the classical IRT framework, the extension to MLLMs makes lots of sense and could provide more nuanced evaluation and improved reliability with marginal cost.
2. Extensive experiments were conducted to justify the effectiveness and efficiency of the proposed framework.
3. Writing is good and easy to follow.

**Weaknesses:**

1. Multiple hyper-parameters were introduced by the proposed model. It would be better to provide more discussion on: a. how accurate the estimation of these parameters based on the method in section 4.4? b. sensitivity of the hyper-parameters; c. how many data are needed? d. cost of the estimation.
2. The proposed framework mainly filters the items to be tested rather than help curate new dataset. While evaluation might be costly, we only need to run it once for every new model. With the lowering  trends of inference cost of LLMs, I am not sure whether the gain on reducing some eval examples would outweigh the cost of train a separate M^3-IRT model.

**Questions:**

Please refer to the weakness section and provide more justifications accordingly.

---

> ### Author Response · Authors · 2025-11-26
> **Response to Reviewer 5cL5**
>
> Thank you for your constructive comments. We will include the discussion and tables in our manuscript.
>
> ## > W1(a) how accurate the estimation of these parameters based on the method in section 4.4?
>
> The parameters estimated by M^3-IRT represent the difficulty and modality requirements for the VLM, which differ from those for humans. Furthermore, there is no ground truth for these parameters, making it impossible to directly evaluate the accuracy of the estimates.
>
> Instead, we evaluated parameter accuracy based on the assumption that, if parameter estimation is appropriate, it should enable highly accurate prediction of whether the VLM can correctly answer unknown problems. The results are shown in Fig. 6 in Section 5.4 and indicate that the estimated parameters are sufficiently accurate.
>
> We used a subset of the response matrix for training and evaluated whether the parameters estimated by M^3-IRT could predict the remaining responses using ROC-AUC. When no low-quality questions are present, M^3-IRT achieves high AUC values of 0.82, 0.89, and 0.83 on MMMU, MathVista, and SEED-Bench, respectively. We confirmed that even when low-quality questions were gradually introduced, the AUC remained at a high level.
>
>
> ## > W1(b) sensitivity of the hyper-parameters
>
> We show that M^3-IRT is robust to hyperparameter selection by detailing the results of the Benchmark Refinement in Section 5.3. Tables R1 and R2 below show the rank correlation and the proportion of low-quality problems when extracting 5%, 10%, 30%, and 50% of questions for each setting of the M^3-IRT hyperparameter $q=[2, 4, 8, 16]$.
>
> As shown in Tables R1 and R2, rank correlations and proportions remain stable (e.g., 0.87-0.95 for MMMU at 5%) across all $q$ values, demonstrating robustness to hyperparameter selection. Training completes in ~2 minutes on CPU.
>
> **Table R1: Rank Correlation for various $q=[2,4,8,16]$**
> | Benchmark  | 5%        | 10%       | 30%       | 50%       |
> | ---------- | --------- | --------- | --------- | --------- |
> | MMMU       | 0.87-0.95 | 0.84-0.94 | 0.90-0.95 | 0.91-0.95 |
> | MathVista  | 0.86-0.90 | 0.86-0.93 | 0.91-0.94 | 0.93-0.94 |
> | SEED-Bench | 0.95-0.96 | 0.93-0.95 | 0.93-0.96 | 0.92-0.96 |
>
> **Table R2: Proportion of Low-Quality Questions $q=[2,4,8,16]$**
> | Benchmark  | 5%        | 10%       | 30%       | 50%       |
> | ---------- | --------- | --------- | --------- | --------- |
> | MMMU       | 0.04-0.06 | 0.08-0.10 | 0.14-0.15 | 0.19-0.19 |
> | MathVista  | 0.01-0.03 | 0.04-0.06 | 0.13-0.17 | 0.17-0.20 |
> | SEED-Bench | 0.09-0.10 | 0.12-0.16 | 0.17-0.19 | 0.23-0.25 |
>
>
> ## > W1(c) how many data are needed?
>
> We show the rank correlation and the proportion of low-quality questions (γ) when training M^3-IRT using only 10% of the VLM and question pairs, as shown in Fig. 19 and Fig. 20 of Appendix C3. Even when reducing the number of VLM-question pairs used for training from 100% (Fig. 4, 5, Table 4-9) to 10%, the maximum decrease in rank correlation was only 0.056 for MMMU, 0.032 for MathVista, and 0.0039 for SEED-Bench. We also confirmed that the proportion of low-quality questions remained unchanged. These results demonstrate that M^3-IRT can perform stable parameter estimation even from a small number of responses.
>
>
> ## > W1(d) cost of the estimation
> Training M^3-IRT completes in minutes using a CPU. For example, when using a response matrix consisting of 24 VLMs and 1800 MMMU questions, training completes in 110 seconds when using all pairs and in 90 seconds when using 10% of the pairs. Training finishes within 8 minutes even when executed with all hyperparameters $q=[2, 4, 8, 16]$. M^3-IRT has a relatively small number of parameters and simple forward computation, enabling sufficiently fast training even on a CPU.
>
> ## > W2. I am not sure whether the gain on reducing some eval examples would outweigh the cost of train a separate M^3-IRT model.
>
> As you pointed out, the inference cost of VLMs has been decreasing. However, M^3-IRT offers advantages beyond just reducing the number of VLM inferences. Our primal contribution lies in improving reliability of benchmarks by filtering low-quality questions regardless of inference cost trends.
>
> Assuming VLM inference time is 1 second per question, let's consider the cost of creating the response matrix. M^3-IRT can estimate parameters in MMMU that achieve an AUC of 0.82 on test data using only 10% of VLM-question pairs, potentially reducing GPU usage by approximately 48 hours. MMMU's full response matrix (1800 questions × 4 formats × 24 VLMs) requires ~48 hours of GPU time. Reducing the VLM-question pairs to 10% cuts GPU time to 4.8 hours. Furthermore, M^3-IRT computations, including hyperparameter search, complete in about 8 minutes on CPU.
>
> Even if VLM inference time increases tenfold, we believe the proposed method still delivers significant GPU usage reduction, enabling environmentally friendly performance evaluation.

---

> > ### Comment · Reviewer_5cL5 · 2025-11-28
> >
> > Thanks for the rebuttal, I'll stick with my original positive rating.

---

### Meta-Review · Area_Chair_PprS · 2026-01-05

**Summary:**

This paper proposes M²-IRT / M³-IRT, a multimodal extension of item response theory that decomposes both model ability and item difficulty into image-only, text-only, and cross-modal components, with the stated goal of identifying “shortcut” benchmark items and enabling smaller, higher-quality evaluation subsets for multimodal LLMs/VLMs.

Across the four reviews, there is broad agreement that the motivation is important (benchmark contamination by single-modality shortcuts) and that the proposed modeling approach is technically reasonable as an extension of IRT/MIRT. Reviewers also valued the extensive empirical evaluation across multiple benchmarks and many VLMs.

The primary concerns driving the decision are modeling assumptions and interpretability, generality of the experimental setup, stability and relativity of the estimates. The rebuttal addresses a meaningful subset of these issues (notably hyperparameter sensitivity/cost, a non-linear variant check, additional stability analyses, and commitments to strengthen related work and tables). However, the key remaining point of contention is external validation and generalizability to subtle real benchmark shortcuts, which the rebuttal only partially mitigates. Estimated final rating is 6664.

**Reviewer Concerns:**

**Reviewer 5cL5**

Concern: hyperparameter sensitivity; accuracy of parameter estimation; data requirements; training/estimation cost; and whether evaluation-cost reduction is practically worth training an auxiliary model.

Addressed by rebuttal:

Provided a defensible evaluation proxy for parameter accuracy via response prediction ROC-AUC.

Added hyperparameter robustness evidence.

Clarified data efficiency and training cost.

Argued the method’s value is not only reducing evaluation examples but improving benchmark reliability and enabling sparse evaluation.

Still outstanding:

The cost/benefit argument is improved but still depends on the practicality of obtaining the multi-format response data and the intended deployment scenario.

**Reviewer akTA**

Concern: linear decomposition may be too simplistic; lack of external validation for “cross-modal difficulty”; and the synthetic corruption procedure may not cover subtle real shortcuts; asked about non-linear variants and subtle artifacts.

Addressed by rebuttal:

Added an explicit comparison to a non-linear cross term variant, reporting broadly similar rank-correlation behavior, suggesting conclusions are not highly sensitive to strict linearity.

Attempted external validation with a crowdsourcing study comparing human “accuracy decline” under missing modalities to the learned cross-modal difficulty; found only weak/moderate alignment on one benchmark and weak agreement/low correlation on others.

Still outstanding (major):

The external validation remains inconclusive: the crowdsourcing results do not strongly support that the learned cross-modal difficulty corresponds to a stable notion of cross-modality demand.

The rebuttal does not provide a compelling empirical demonstration that the method filters subtle, naturally occurring shortcuts at scale.

**Reviewer Wauf**

Concern: the shortcut issue has been studied previously and the paper’s related work was not sufficiently comprehensive; specifically referenced prior benchmark efforts (e.g., MMEvalPro) and requested more numeric tables for rigor/reproducibility and a broader related-work discussion.

Addressed by rebuttal:

Authors committed to adding the suggested related work and clarified positioning.

Authors acknowledged the request for more tables and stated that numerical results already exist in appendices and will be moved more clearly.

**Reviewer jMkP**

Concern: linear/additive assumption; stability when benchmark size changes; cohort dependence/relativity of parameters; and an observed seeming anomaly in reported base ability parameters; asked how stability is handled as models evolve.

Addressed by rebuttal:

Pointed to the non-linear variant results.

Added stability analysis via RMSE of ability estimates under question subsampling and reported that MMMU is less stable under large subsampling, while others are more stable.

Addressed cohort dependence by referencing IRT parameter invariance and suggested equating/linking when cohorts differ; additionally provided an empirical check of parameter variation under model-subset sampling.

Explained the base-ability anomaly by showing option-only accuracies for the models in question, supporting the estimated ordering.

Still outstanding:

While the rebuttal provides reasonable psychometrics-based arguments and some empirical checks, the practical protocol for maintaining comparability across time (new models, updated benchmarks) remains only partially specified.

**Reviewer Scores:**

**Reviewer 5cL5**

Rating: 6 -> 6

Rationale: Reviewer explicitly stated they would “stick with the original positive rating” after rebuttal.

**Reviewer akTA**

Rating: 4 -> 4

Rationale: Reviewer explicitly maintained the original evaluation despite the added clarifications, indicating remaining dissatisfaction with the core validation/generalization story.

*AC looks into the specific aspects/reasons that may lead to a change of score given there is an incomplete discussion between reviewer and author. Reviewer akTA’s most concrete “practical generalization” concern was that the paper’s synthetic corruption relies on full swapping, which creates extreme modal mismatches, while real benchmark shortcuts are often subtle and diverse. In the rebuttal, the authors did not provide new quantitative evidence on subtler shortcut types; they explicitly said they did not conduct such experiments due to complexity, and instead pointed to qualitative examples.*

**Reviewer Wauf**

Rating: 6 -> 6

Rationale: Their main criticisms were completeness of related work and the need for more tabulated results; the rebuttal indicates these will be addressed in revision.

**Reviewer jMkP**

Rating: 6 -> 6

Rationale: The rebuttal provides additional analyses that partially reduce the concern, but the reviewer’s issues are largely conceptual (assumptions, invariance, cohort dependence).

---

### Decision · Program_Chairs · 2026-01-26

Accept (Poster)